# A Theoretical Analysis of Optimistic Proximal Policy Optimization in Linear Markov Decision Processes

**Han Zhong**
Center for Data Science
Peking University
hanzhong@stu.pku.edu.cn

**Tong Zhang**
Department of Mathematics
HKUST
tongzhang@tongzhang-ml.org

## Abstract

The proximal policy optimization (PPO) algorithm stands as one of the most prosperous methods in the field of reinforcement learning (RL). Despite its success, the theoretical understanding of PPO remains deficient. Specifically, it is unclear whether PPO or its optimistic variants can effectively solve linear Markov decision processes (MDPs), which are arguably the simplest models in RL with function approximation. To bridge this gap, we propose an optimistic variant of PPO for episodic adversarial linear MDPs with full-information feedback, and establish a $\widetilde{\mathcal{O}}(d^{3/4}H^2K^{3/4})$ regret for it. Here $d$ is the ambient dimension of linear MDPs, $H$ is the length of each episode, and $K$ is the number of episodes. Compared with existing policy-based algorithms, we achieve the state-of-the-art regret bound in both stochastic linear MDPs and adversarial linear MDPs with full information. Additionally, our algorithm design features a novel multi-batched updating mechanism and the theoretical analysis utilizes a new covering number argument of value and policy classes, which might be of independent interest.

## 1 Introduction

Reinforcement learning (RL) [34] is a prominent approach to solving sequential decision making problems. Its tremendous successes [19, 32, 33, 5] can be attributed, in large part, to the advent of deep learning [22] and the development of powerful deep RL algorithms [24, 28, 29, 11]. Among these algorithms, the proximal policy optimization (PPO) [29] stands out as a particularly significant approach. Indeed, it continues to play a pivotal role in recent advancements in large language models [27].

Motivated by the remarkable empirical success of PPO, numerous studies seek to provide theoretical justification for its effectiveness. In particular, Cai et al. [6] develop an optimistic variant of the PPO algorithm in adversarial linear mixture MDPs with full-information feedback [4, 25], where the transition kernel is a linear combination of several base models. Theoretically, they show that the optimistic variant of PPO is capable of tackling problems with large state spaces by establishing a sublinear regret that is independent of the size of the state space. Building upon this work, He et al. [14] study the same setting and refine the regret bound derived by Cai et al. [6] using the weighted regression technique [44]. However, the algorithms in Cai et al. [6], He et al. [14] and other algorithms for linear mixture MDPs [4, 44] are implemented in a model-based manner and require an integration of the individual base model, which can be computationally expensive or even intractable in general. Another arguably simplest RL model involving function approximation is linear MDP [38, 17], which assumes that the reward functions and transition kernel enjoy a low-rank representation. For this model, several works [17, 15, 3, 13] propose value-based algorithms that directly approximate the value function and provide regret guarantees. To demonstrate the efficiency of PPO in linear MDPs from a theoretical perspective, one potential approach is to extend the results

37th Conference on Neural Information Processing Systems (NeurIPS 2023).

Table 1: A comparison with closely related works on policy optimization for linear MDPs. Here "Sto." and "Adv." represent stochastic rewards and adversarial rewards, respectively. Additionally, "Bandit" and "Full-infor." signify bandit feedback and full-information feedback. We remark that Zanette et al. [40] do not consider the regret minimization problem and the regret reported in the table is implied by their sample complexity. We will compare the sample complexity provided in Zanette et al. [40] and the complexity implied by our regret in Remark 3.2.

| | Sto. + Bandit | Adv. + Full-infor. | Adv. + Bandit | Regret |
|---|---|---|---|---|
| [40] | ✓ | ✗ | ✗ | $\widetilde{\mathcal{O}}(d^{3/4}H^{13/4}K^{3/4})$ |
| [8] | ✓ | ✓ | ✓ | $\widetilde{\mathcal{O}}(d^{2/3}A^{1/9}H^{20/9}K^{8/9})$ |
| [31] | ✓ | ✓ | ✓ | $\widetilde{\mathcal{O}}(dH^2K^{6/7})$ |
| Our Work | ✓ | ✓ | ✗ | $\widetilde{\mathcal{O}}(d^{3/4}H^2K^{3/4})$ |

of Cai et al. [6], He et al. [14] to linear MDPs. However, this extension poses significant challenges due to certain technical issues that are unique to linear MDPs. See §1.1 for a detailed description.

In this paper, we address this technical challenge and prove that the optimistic variant of PPO is provably efficient for stochastic linear MDPs and even adversarial linear MDPs with full-information feedback. Our contributions are summarized below.

- In terms of algorithm design, we propose a new algorithm OPPO+ (Algorithm 1), an optimistic variant of PPO, for adversarial linear MDPs with full-information feedback. Our algorithm features two novel algorithm designs including a *multi-batched updating mechanism* and a *policy evaluation step via average rewards*.

- Theoretically, we establish a $\widetilde{\mathcal{O}}(d^{3/4}H^2K^{3/4})$ regret for OPPO+, where $d$ is the ambient dimension of linear MDPs, $H$ is the horizon, and $K$ is the number of episodes. To achieve this result, we employ two new techniques. Firstly, we adopt a novel covering number argument for the value and policy classes, as explicated in §C. Secondly, in Lemma 4.3, we meticulously analyze the drift between adjacent policies to control the error arising from the policy evaluation step using average rewards.

- Compared with existing policy optimization algorithms, our algorithm achieves a better regret guarantee for both stochastic linear MDPs and adversarial linear MDPs with full-information feedback (to our best knowledge). See Table 1 for a detailed comparison.

In summary, our work provides a new theoretical justification for PPO in linear MDPs. To illustrate our theory more, we highlight the challenges and our novelties in §1.1.

## 1.1 Challenges and Our Novelties

**Challenge 1: Covering Number of Value Function Class.** In the analysis of linear MDPs (see Lemma B.3 or Lemma B.3 in Jin et al. [17]), we need to calculate the covering number of $\mathcal{V}_h^k$, which is the function class of the estimated value function at the $h$-th step of the $k$-th episode and takes the following form:

$$\mathcal{V}_h^k = \{V(\cdot) = \langle Q_h^k(\cdot, \cdot), \pi_h^k(\cdot \mid \cdot) \rangle_{\mathcal{A}}\},$$

where $Q_h^k$ and $\pi_h^k$ are estimated Q-function and policy at the $h$-th step of the $k$-th episode, respectively. For value-based algorithms (e.g., LSVI-UCB in Jin et al. [17]), $\pi_h^k$ is the greedy policy with respect to $Q_h^k$. Then we have

$$\mathcal{V}_h^k = \{V(\cdot) = \max_a Q_h^k(\cdot, a)\}.$$

Since $\max_a$ is a contraction map, it suffices to calculate $\mathcal{N}(\mathcal{Q}_h^k)$, where $\mathcal{Q}_h^k$ is the function class of $Q_h^k$ and $\mathcal{N}(\cdot)$ denotes the covering number. By a standard covering number argument (Lemma D.4 or Lemma D.6 in Jin et al. [17]), we can show that $\log \mathcal{N}(\mathcal{Q}_h^k) \leq \widetilde{\mathcal{O}}(d^2)$. However, for policy-based algorithms such as PPO, $\pi_h^k$ is a stochastic policy, which makes the log-covering number $\log \mathcal{N}(\mathcal{V}_h^k)$ may have a polynomial dependency on the size of action space $\mathcal{A}$ (log-covering number

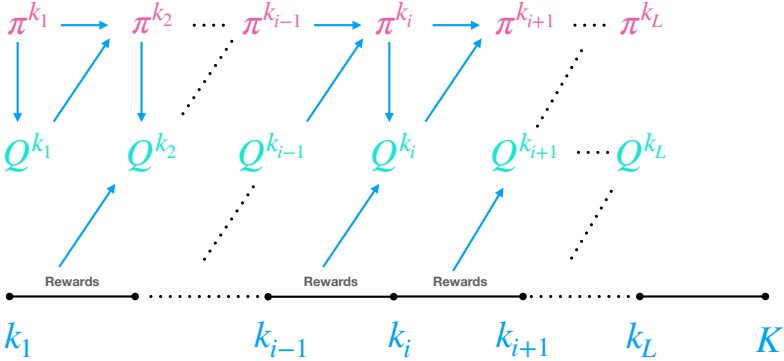

Figure 1: Algorithm diagram. The entire $K$ episodes are partitioned into $L = K/B$ batches, with each batch containing $B$ consecutive episodes. The policy/value updating only occurs at the beginning of each batch. The policy evaluation uses the reward function in the last batch.

of $|\mathcal{A}|$-dimensional probability distributions is at the order of $|\mathcal{A}|$). We also remark that linear mixture MDPs are more amenable to theoretical analysis compared to linear MDPs, as they do not necessitate the calculation of the covering number of $\mathcal{V}_h^k$ [4, 6]. As a result, the proof presented by Cai et al. [6] is valid for linear mixture MDPs, but cannot be extended to linear MDPs. See §A for more elaboration of technical challenges.

**Novelty 1: Multi-batched Updating and A New Covering Number Argument.** Our key observation is that if we improve the policy like PPO (see (3.2) or Schulman et al. [29], Cai et al. [6]), $\pi_h^k$ admits a softmax form, i.e.,

$$\pi_h^k(\cdot \mid \cdot) \propto \exp\Big(\sum_{i=1}^{l} Q_h^{k_i}(\cdot, \cdot)\Big).$$

Here $\{k_i\}_{i=1}^l$ is a sequence of episodes, where $k_i \leq k$ for all $i \in [l]$, denoting the episodes in which our algorithm performs policy optimization prior to the $k$-th episode. By a technical lemma (Lemma C.3), we can show that

$$\log \mathcal{N}(\text{class of } \pi_h^k) \lesssim \sum_{i=1}^{l} \log \mathcal{N}(\mathcal{Q}_h^{k_i}) \lesssim \widetilde{\mathcal{O}}(l \cdot d^2).$$

If we perform the policy optimization in each episode like Cai et al. [6], Shani et al. [30], $l$ may linear in $K$ and the final regret bound becomes vacuous. Motivated by this, we use a multi-batched updating scheme. In specific, OPPO+ divides the whole learning process into several batches and only updates policies at the beginning of each batch. See Figure 1 for visualization. For example, if the number of batches is $K^{1/2}$ (i.e., each batch consists of consecutive $K^{1/2}$ episodes), we have $l \leq K^{1/2}$ and the final regret is at the order of $\widetilde{\mathcal{O}}(K^{3/4})$. Here we assume $K^{1/2}$ is a positive integer for simplicity. See §C for details.

**Challenge 2: Adversarial Rewards.** Compared with previous value-based algorithms [e.g., 17], one superiority of optimistic PPO is that they can learn adversarial MDPs with full-information feedback [6, 30]. In Cai et al. [6], Shani et al. [30], the policy evaluation step is that

$$Q_h^k = r_h^k + \widehat{\mathbb{P}}_h^k V_{h+1}^k + \text{bonus function}, \quad \forall h \in [H], \quad V_{h+1}^k = 0,$$

where $r_h^k$ is the adversarial reward function and $\widehat{\mathbb{P}}_h^k V_{h+1}^k$ is the estimator of the expected next step value of $V_{h+1}^k$. This policy evaluation step is invalid if we use the multi-batched updating. Consider the following case, if the number of batches is $K^{1/2}$ and

$$r_h^k(\cdot, \cdot) = \begin{cases} 0 & k \in \{iK^{1/2} + 1\}_{i=0}^{K^{1/2}} \\ \text{arbitrary} & \text{otherwise} \end{cases}.$$

Then the algorithm only uses zero rewards to find the optimal policy in hindsight with respect to arbitrary adversarial rewards, which is obviously impossible.

**Novelty 2: Policy Evaluation via Average Reward and Smoothness Analysis.** To tackle the above challenge, we adopt the following policy evaluation step at the beginning of each batch

$$Q_h^k = \bar{r}_h^k + \widehat{\mathbb{P}}_h^k V_{h+1}^k + \text{bonus function}, \quad \forall h \in [H], \quad V_{H+1}^k = 0,$$

where $\bar{r}_h^k$ the average reward of the last batch:

$$\bar{r}_h^k = \frac{\sum (\text{reward functions of the last batch})}{\text{batch size}}.$$

Let $\pi^{k_i}$ denote the policy executed in the $i$-th batch. Intuitively, $\pi_{k_{i+1}}$ is the desired policy within $i$-th batch since its calculation only uses the rewards in the first $(i-1)$ batches (cf. Figure 1). Hence, compared with Cai et al. [6], Shani et al. [30], we need to handle the gap between the performance of $\pi^{k_{i+1}}$ and $\pi^{k_i}$ in the $i$-th batch. Fortunately, this error can be controlled due to the "smoothness" of policies in adjacent batches. See Lemma 4.3 for details.

## 1.2 Related Works

**Policy Optimization Algorithms.** The seminal work of Schulman et al. [29] proposes the PPO algorithm, and a line of following works seeks to provide theoretical guarantees for it. In particular, Cai et al. [6] proposes the optimistic PPO (OPPO) algorithm for adversarial linear mixture MDPs and establishes regret $\widetilde{\mathcal{O}}(d\sqrt{H^4 K})$ for it. Then, He et al. [14] improve the regret to $\widetilde{\mathcal{O}}(d\sqrt{H^3 K})$ by the weighted regression technique [44]. Besides their works on linear mixture MDPs, Shani et al. [30], Wu et al. [37] provide fine-grained analysis of optimistic variants of PPO in the tabular case. The works of Fei et al. [9], Zhong et al. [42] show that optimistic variants of PPO can solve non-stationary MDPs. However, none of these works show that the optimistic variant of PPO is provably efficient for linear MDPs.

There is another line of works [2, 10, 40] proposes optimistic policy optimization algorithms based on the natural policy gradient (NPG) algorithm [18] and the policy-cover technique. But their works are limited to the stochastic linear MDPs, while our work can tackle adversarial rewards. Compared with their results for stochastic linear MDPs, our work can achieve a better regret and compatible sample complexity. See Table 1 and Remark 3.2 for a detailed comparison. Several recent works [26, 23, 20, 8, 31] study the more challenging problem of learning adversarial linear MDPs with only bandit feedback, which is beyond the scope of our work. Without access to exploratory policies or even known transitions, their regret is at least $\widetilde{\mathcal{O}}(K^{6/7})$ [31], while our work achieves a better $\widetilde{\mathcal{O}}(K^{3/4})$ regret with the full-information feedback assumption.

**RL with Linear Function Approximation.** Our work is related to previous works proposing value-based algorithms for linear MDPs [38, 17]. The work of Yang and Wang [38] develops the first sample efficient algorithm for linear MDPs with a generative model. Then Jin et al. [17] proposes the first provably efficient algorithms for linear MDPs in the online setting. The results of Jin et al. [17] are later improved by [35, 15, 13, 3]. In particular, Agarwal et al. [3], He et al. [13] show that the nearly minimax optimal regret $\widetilde{\mathcal{O}}(d\sqrt{H^3 K})$ is achievable in stochastic linear MDPs. Compared with these value-based algorithms, our work can tackle the more challenging adversarial linear MDPs with full-information feedback.

There is another line of works [4, 25, 6, 41, 14, 45, 44, 43] studying linear mixture MDPs, which is another model of RL with linear function approximation. It can be shown that linear MDPs and linear mixture MDPs are incompatible in the sense that neither model is a special case of the other. Among these works, Zhou et al. [44], Zhou and Gu [43] establishes nearly minimax regret $\widetilde{\mathcal{O}}(d\sqrt{H^3 K})$ for stochastic linear mixture MDPs. Our work is more related to Cai et al. [6], He et al. [14] on adversarial linear mixture MDPs with full-information feedback. We have remarked that it is nontrivial to extend their results to linear MDPs.

## 2 Preliminaries

**Notations.** We use $\mathbb{N}^+$ to denote the set of positive integers. For any $H \in \mathbb{N}^+$, we denote $[H] = \{1, 2, \ldots, H\}$. For any $H \in \mathbb{N}^+$ and $x \in \mathbb{R}$, we use the notation $\min\{x, H\}^+ =$

$\min\{H, \max\{0, x\}\}$. Besides, we denote by $\Delta(\mathcal{A})$ the set of probability distributions on the set $\mathcal{A}$. For any two distributions $P$ and $Q$ over $\mathcal{A}$, we denote $\mathrm{KL}(P\|Q) = \mathbb{E}_{a \sim P}[\log \mathrm{d}P(a)/\mathrm{d}Q(a)]$.

**Episodic Adversarial MDPs.** We consider an episodic MDP $\mathcal{M}$, which is denoted by a tuple

$$(\mathcal{S}, \mathcal{A}, H, K, \{r_h^k\}_{(k,h) \in [K] \times [H]}, \mathcal{P} = \{\mathcal{P}_h\}_{h \in [H]}),$$

where $\mathcal{S}$ is the state space, $\mathcal{A}$ is the action space, $H$ is the length of each episode, $K$ is the number of episodes, $r_h^k : \mathcal{S} \times \mathcal{A} \mapsto [0, 1]$ is the deterministic[1] reward function at the $h$-th step of $k$-th episode, $\mathcal{P}_h$ is the transition kernel with $\mathcal{P}_h(s' \mid s, a)$ being the transition probability for state $s$ to transfer to the next state $s'$ given action $a$ at the $h$-th step. We consider the *adversarial* MDPs with *full-information feedback*, which means that the reward $\{r_h^k\}_{h \in [H]}$ is adversarially chosen by the environment at the beginning of the $k$-th episode and revealed to the learner after the $k$-th episode.

A policy $\pi = \{\pi_h\}_{h \in [H]}$ is a collection of $H$ functions, where $\pi_h : \mathcal{S} \mapsto \Delta(\mathcal{A})$ is a function that maps a state to a distribution over action space at step $h$. For any policy $\pi$ and reward function $\{r_h^k\}_{h \in [H]}$, we define the value function $V_h^{\pi,k} : \mathcal{S} \mapsto \mathbb{R}$ and Q-function $Q_h^{\pi,k} : \mathcal{S} \times \mathcal{A} \mapsto \mathbb{R}$ as

$$V_h^{\pi,k}(x) = \mathbb{E}_\pi\bigg[ \sum_{h'=h}^{H} r_{h'}^k(x_{h'}, a_{h'}) \bigg| x_h = x \bigg],$$

$$Q_h^{\pi,k}(x, a) = \mathbb{E}_\pi\bigg[ \sum_{h'=h}^{H} r_{h'}^k(x_{h'}, a_{h'}) \bigg| x_h = x, a_h = a \bigg],$$

for any $(x, a, k, h) \in \mathcal{S} \times \mathcal{A} \times [K] \times [H]$. Here the expectation $\mathbb{E}_\pi[\cdot]$ is taken with respect to the randomness of the trajectory induced by policy $\pi$ and transition kernel $\mathcal{P}$. It is well-known that the value function and Q-function satisfy the following Bellman equation for any $(x, a) \in \mathcal{S} \times \mathcal{A}$,

$$V_h^{\pi,k}(x) = \langle Q_h^{\pi,k}(x, \cdot), \pi_h(\cdot \mid x) \rangle_{\mathcal{A}}, \quad Q_h^{\pi,k}(x, a) = r_h^k(x, a) + (\mathbb{P}_h V_{h+1}^{\pi,k})(x, a), \tag{2.1}$$

where $\langle \cdot, \cdot \rangle_{\mathcal{A}}$ denotes the inner product over the action space $\mathcal{A}$ and we will omit the subscript when it is clear from the context. Here $\mathbb{P}_h$ is the operator defined as

$$(\mathbb{P}_h V)(x, a) = \mathbb{E}_{x' \sim \mathcal{P}_h(\cdot \mid x, a)}[V(x')], \qquad \forall V : \mathcal{S} \mapsto \mathbb{R}. \tag{2.2}$$

**Interaction Process and Learning Objective.** We consider the online setting, where the learner improves her performance by interacting with the environment repeatedly. The learning process consists of $K$ episodes and each episode starts from a fixed initial state $x_1$[2]. At the beginning of the $k$-th episode, the environment adversarially chooses reward functions $\{r_h^k\}_{h \in [H]}$, which can depend on previous $(k-1)$ trajectories. Then the agent determines a policy $\pi^k$ and receives the initial state $x_1^k = x_1$. At each step $h \in [H]$, the agent receives the state $x_h^k$, chooses an action $a_h^k \sim \pi_h^k(\cdot \mid x_h^k)$, receives the reward function $r_h^k$, and transits to the next state $x_{h+1}^k$. The $k$-th episode ends after $H$ steps.

We evaluate the performance of an online algorithm by the notion of *regret* [7], which is defined as the value difference between the executed policies and the optimal policy in hindsight:

$$\mathrm{Regret}(K) = \max_\pi \sum_{k=1}^{K} \big( V_1^{\pi,k}(x_1) - V_1^{\pi^k,k}(x_1) \big).$$

For simplicity, we denote the optimal policy in hindsight by $\pi^*$, i.e., $\pi^* = \mathrm{argmax}_\pi \sum_{k=1}^{K} V_1^{\pi,k}(x_1)$.

**Linear MDPs.** We focus on the linear MDPs [38, 17], where the transition kernels are linear in a known feature map.

**Definition 2.1** (Linear MDP). We say an MDP $(\mathcal{S}, \mathcal{A}, H, K, \{r_h^k\}_{(k,h) \in [K] \times [H]}, \mathcal{P} = \{\mathcal{P}_h\}_{h \in [H]})$ is a linear MDP if there exists a known feature $\phi : \mathcal{S} \times \mathcal{A} \mapsto \mathbb{R}^d$ such that for any $(x, a, k, h) \in \mathcal{S} \times \mathcal{A} \times [K] \times [H]$, we have

$$\mathcal{P}_h(x' \mid x, a) = \phi(x, a)^\top \mu_h(x'),$$

where $\mu_h = (\mu_h^{(1)}, \dots, \mu_h^{(d)})$ are $d$ unknown signed measures over $\mathcal{S}$ satisfying $\|\mu_h(\mathcal{S})\|_2 \le \sqrt{d}$.

---

[1]This assumption is without loss of generality since our subsequent results are ready to be extended to the stochastic reward case.

[2]Our subsequent analysis can be generalized to the case where the initial state is chosen from a fixed distribution across all episodes.

**Algorithm 1** OPPO+

---

**Require:** Batch size $B \in \mathbb{N}^+$, regularization parameter $\lambda > 0$, and confidence radius $\beta > 0$.
1: Initialize $\{Q_h^0\}$, $\{r_h^k\}_{-B \leq k \leq 0}$ as zero functions and $\{\pi_h^0\}$ as uniform distributions on $\mathcal{A}$, $\forall h \in [H]$.
2: Let $L = K/B$, $i = 1$, and $k_i = (i-1) \cdot B + 1$ for $1 \leq i \in [L]$.
3: **for** episode $k = 1, 2, \ldots, K$ **do**
4:     Receive the initial state $x_1^k$.
5:     **if** $k = k_i$ **then**
6:         $V_{H+1}^k(\cdot) \leftarrow 0$.
7:         **for** step $h = 1, 2, \ldots, H$ **do**
8:             Update the policy by $\pi_h^k(\cdot \mid \cdot) \propto \pi_h^{k-1}(\cdot \mid \cdot) \cdot \exp\{\alpha \cdot Q_h^{k-1}(\cdot, \cdot)\}$.
9:         **end for**
10:      **for** step $h = H, H-1, \ldots, 1$ **do**
11:         $\Lambda_h^k \leftarrow \sum_{\tau=1}^{k-1} \phi(x_h^\tau, a_h^\tau) \phi(x_h^\tau, a_h^\tau)^\top + \lambda \cdot I_d$.
12:         $w_h^k \leftarrow (\Lambda_h^k)^{-1} \sum_{\tau=1}^{k-1} \phi(x_h^\tau, a_h^\tau) \cdot V_{h+1}^k(x_{h+1}^\tau)$.
13:         $\bar{r}_h^k(\cdot, \cdot) \leftarrow (\sum_{j=k_{i-1}}^{k_i-1} r_h^j(\cdot, \cdot))/B$.
14:         $\Gamma_h^k(\cdot, \cdot) \leftarrow \beta \cdot [\phi(\cdot, \cdot)^\top (\Lambda_h^k)^{-1} \phi(\cdot, \cdot)]^{1/2}$.
15:         $\widehat{\mathbb{P}}_h^k V_{h+1}^k(\cdot, \cdot) \leftarrow \min\{\phi(\cdot, \cdot)^\top w_h^k + \Gamma_h^k(\cdot, \cdot), H - h\}^+$.
16:         $Q_h^k(\cdot, \cdot) \leftarrow \bar{r}_h^k(\cdot, \cdot) + \widehat{\mathbb{P}}_h^k V_{h+1}^k(\cdot, \cdot)$.
17:         $V_h^k(\cdot) \leftarrow \langle Q_h^k(\cdot, \cdot), \pi_h^k(\cdot \mid \cdot) \rangle_{\mathcal{A}}$.
18:      **end for**
19:      $i \leftarrow i + 1$
20:     **else**
21:         $Q_h^k \leftarrow Q_h^{k-1}, V_h^k \leftarrow V_h^{k-1}, \pi_h^k \leftarrow \pi_h^{k-1}, \bar{r}_h^k \leftarrow \bar{r}_h^{k-1} \; \forall h \in [H]$.
22:     **end if**
23:     **for** $h = 1, 2, \ldots, H$ **do**
24:         Take the action following $a_h^k \sim \pi_h^k(\cdot \mid x_h^k)$.
25:         Observe the reward function $r_h^k(\cdot, \cdot)$ and receive the next state $x_{h+1}^k$.
26:     **end for**
27: **end for**

---

Since we have access to the full-information feedback, we do not assume the reward functions are linear in the feature map $\phi$ like Jin et al. [17]. We also remark that the adversarial linear MDP with full-information feedback is a more challenging problem than the stochastic linear MDP with bandit feedback studied in Jin et al. [17]. In fact, for stochastic linear MDPs, we can assume the reward functions are known without loss of generality since learning the linear transition kernel is more difficult than the linear reward.

## 3 Algorithm

In this section, we propose a new algorithm OPPO+ (Algorithm 1) to solve adversarial linear MDPs with full-information feedback. In what follows, we highlight the key steps of OPPO+.

**Multi-batched Updating.** Due to the technical issue elaborated in §1.1, we adopt the multi-batched updating rule. In specific, OPPO+ divides the total $K$ episodes into $L = K/B$ batches and each batch consists of $B$ consecutive episodes. Here we assume $K/B$ is a positive integer without loss of generality[3]. For ease of presentation, we use $k_i = (i-1) \cdot B + 1$ to denote the first episode in the $i$-th batch. When the $k$-th episode is the beginning of a batch (i.e, $k = k_i$ for some $i \in [L]$), OPPO+ performs the following *policy improvement step* and *policy evaluation step*.

**Policy Improvement.** In the policy improvement step of the $k$-th episode ($k = k_i$ for some $i \in [L]$), OPPO+ calculates $\pi^k$ based on the previous policy $\pi^{k-1}$ using PPO [29]. In specific, OPPO+ updates

---

[3]We can only consider the first $B \cdot \lfloor K/B \rfloor$ episodes since the remaining episodes will lead at most $BH$ regret, which is a non-dominant term in final regret bound

$\pi^k$ by solving the following proximal policy optimization problem:

$$\pi^k \leftarrow \operatorname*{argmax}_{\pi} \left\{ L_{k-1}(\pi) - \alpha^{-1} \cdot \mathbb{E}_{\pi^{k-1}} \left[ \sum_{h=1}^{H} \mathrm{KL}\big( \pi_h(\cdot \mid x_h) \| \pi_h^{k-1}(\cdot \mid x_h) \big) \right] \right\}, \qquad (3.1)$$

where $\alpha > 0$ is the stepsize that will be specified in Theorem 3.1, and $L_{k-1}(\pi)$ takes form

$$L_{k-1}(\pi) = V_1^{\pi^{k-1},k-1}(x_1^k) + \mathbb{E}_{\pi^{k-1}} \left[ \sum_{h=1}^{H} \langle Q_h^{k-1}(x_h, \cdot), \pi_h(\cdot \mid x_h) - \pi_h^{k-1}(\cdot \mid x_h) \rangle \right],$$

which is proportional to the local linear function of $V_1^{\pi,k-1}(x_1^k)$ at $\pi^{k-1}$ and replaces the unknown Q-function $Q_h^{\pi^{k-1},k-1}$ by the estimated one $Q_h^{k-1}$ for any $h \in [H]$. It is not difficult to show that the updated policy $\pi^k$ obtained in (3.1) admits the following closed form:

$$\pi_h^k(\cdot \mid x) \propto \pi_h^{k-1}(\cdot \mid x) \cdot \exp\big(\alpha \cdot Q_h^{k-1}(x, \cdot)\big), \qquad \forall (x, h) \in \mathcal{S} \times [H]. \qquad (3.2)$$

**Policy Evaluation.** In the policy evaluation step of the $k$-th episode ($k = k_i$ for some $i \in [K]$), OPPO+ lets $V_{H+1}^k$ be the zero function and iteratively calculates the estimated Q-function $\{Q_h^k\}_{h\in[H]}$ in the order of $h = H, H-1, \ldots, 1$. Now we present the policy evaluation at the $h$-th step given estimated value $V_{h+1}^k$. By the definitions of linear MDP in Definition 2.1 and the operator $\mathbb{P}_h$ in (2.2), we know $\mathbb{P}_h V_{h+1}^k$ is linear in the feature map $\phi$. Inspired by this, we estimate its linear coefficient by solving the following ridge regression:

$$w_h^k = \operatorname*{argmin}_{w \in \mathbb{R}^d} \sum_{\tau=1}^{k-1} \big( \phi(x_h^\tau, a_h^\tau)^\top w - V_{h+1}^k(x_{h+1}^\tau) \big)^2 + \lambda \cdot I_d, \qquad (3.3)$$

where $\lambda > 0$ is the regularization parameter and $I_d$ is the identify matrix. By solving (3.3), we have

$$w_h^k = (\Lambda_h^k)^{-1} \Big( \sum_{\tau=1}^{k-1} \phi(x_h^\tau, a_h^\tau) \cdot V_{h+1}^k(x_{h+1}^\tau) \Big), \quad \text{where } \Lambda_h^k = \sum_{\tau=1}^{k-1} \phi(x_h^\tau, a_h^\tau) \phi(x_h^\tau, a_h^\tau)^\top + \lambda \cdot I_d.$$

Based on this linear coefficient, we construct the estimator $\widehat{\mathbb{P}}_h^k V_{h+1}^k$ as

$$(\widehat{\mathbb{P}}_h^k V_{h+1}^k)(\cdot, \cdot) = \min\{\phi(\cdot, \cdot)^\top w_h^k + \Gamma_h^k(\cdot, \cdot), H - h\}^+,$$

where $\Gamma_h^k = \beta \cdot (\phi(\cdot, \cdot)^\top (\Lambda_h^k)^{-1} \phi(\cdot, \cdot))^{1/2}$ is the bonus function and $\beta > 0$ is a parameter that will be specified in Theorem 3.1. This form of bonus function also appears in the literature on linear bandits [21] and linear MDPs [17]. Finally, we update $Q_h^k$ and $V_h^k$ by

$$Q_h^k(\cdot, \cdot) = \bar{r}_h^k(\cdot, \cdot) + (\widehat{\mathbb{P}}_h^k V_{h+1}^k)(\cdot, \cdot), \qquad V_h^k(\cdot) = \langle Q_h^k(\cdot, \cdot), \pi_h^k(\cdot \mid \cdot) \rangle_{\mathcal{A}}. \qquad (3.4)$$

Here $\bar{r}_h^k$ is the average reward function in the last batch, that is

$$\bar{r}_h^k(\cdot, \cdot) = \frac{\sum_{j=k_{i-1}}^{k_i - 1} r_h^j(\cdot, \cdot)}{B}, \qquad (3.5)$$

where $k = k_i = (i-1) \cdot B + 1$ and $k_{i-1} = (i-2) \cdot B + 1$.

Here we would like to make some comparisons between our algorithm OPPO+ and other related algorithms. Different from previous value-based algorithms that take simply take the greedy policy with respect to the estimated Q-functions [17], OPPO+ involves a policy improvement step like PPO [29]. This step is key to tackling adversarial rewards. The most related algorithm is OPPO proposed by Cai et al. [6], which performs the policy optimization in linear mixture MDPs. The main difference between OPPO+ and OPPO is that we introduce a multi-batched updating and an average reward policy evaluation, which are important for solving linear MDPs (cf. §1.1). Finally, we remark that the multi-batched updating scheme is adopted by previous work on bandits [12] and RL [36]. But their algorithms are value-based and cannot tackle adversarial rewards.

**Theorem 3.1** (Regret). Fix $\delta \in (0,1]$ and $K \geq d^3$. Let $B = \sqrt{d^3 K}$ and $\alpha = \sqrt{2B \log |\mathcal{A}|/(KH^2)}$, $\lambda = 1$, $\beta = \mathcal{O}(d^{1/4} H K^{1/4} \iota^{1/2})$ with $\iota = \log(dHK|\mathcal{A}|/\delta)$, then with probability at least $1 - \delta$, the regret of Algorithm 1 satisfies

$$\mathrm{Regret}(K) \leq \mathcal{O}(d^{3/4} H^2 K^{3/4} \log |\mathcal{A}| \cdot \iota) + \mathcal{O}(d^{5/2} H^2 K^{1/2} \cdot \iota).$$

*Proof.* By See §4 for a detailed proof. □

To illustrate our theory more, we make several remarks as follows.

**Remark 3.2** (Sample Complexity). Since learning adversarial linear MDPs with full-information feedback is more challenging than learning stochastic linear MDPs with bandit feedback, the result in Theorem 3.1 also holds for stochastic linear MDPs. By the standard online-to-batch argument [16], we have that Algorithm 1 can find an $\epsilon$-optimal policy using at most

$$\widetilde{\mathcal{O}}\Big(\frac{d^3 H^8}{\epsilon^4} + \frac{d^5 H^4}{\epsilon^2}\Big)$$

samples. Compared with the sample complexity $\widetilde{\mathcal{O}}(d^3 H^{13}/\epsilon^3)$ in Zanette et al. [40], we have a better dependency on $H$ but a worse dependency on $\epsilon$. Moreover, by the standard sample complexity to regret argument [16], their sample complexity only gives a $\widetilde{\mathcal{O}}(d^{3/4} H^{13/4} K^{3/4})$, which is worse than our regret in Theorem 3.1. More importantly, the algorithm in Zanette et al. [40] lacks the ability to handle adversarial rewards, whereas our proposed algorithm overcomes this limitation.

**Remark 3.3** (Optimality of Results). For stochastic linear MDPs, Agarwal et al. [3], He et al. [13] design value-based algorithms with $\widetilde{\mathcal{O}}(d\sqrt{H^3 K})$ regret, which matches the lower bound $\Omega(d\sqrt{H^3 K})$ [44] up to logarithmic factors. It remains unclear whether policy-based based algorithms can achieve the nearly minimax optimal regret and we leave this as future work. For the more challenging adversarial linear MDPs with full-information feedback, we achieve the state-of-the-art regret bound. In this setup, a direct lower bound is $\Omega(d\sqrt{H^3 K})$ [44, 14], and we conjecture this lower bound is tight. It would be interesting to design algorithms with $\sqrt{K}$ regret or even optimal regret in this setting.

**Remark 3.4** (Beyond the Linear Function Approximation). The work of Agarwal et al. [2] extends their results to the kernel function approximation setting. We conjecture that our results can also be extended to RL with kernel and neural function approximation by the techniques in Yang et al. [39].

## 4 Proof of Theorem 3.1

*Proof.* Recall that $B \in \mathbb{N}^+$ is the batch size, $L = K/B$ is the number of batches, and $k_i = (i-1) \cdot B + 1$ for any $i \in [L]$. For any $k \in [K]$, we use $t_k$ to denote the $k_i$ satisfying $k_i \leq k < k_{i+1}$. Moreover, we define the Bellman error as

$$\delta_h^k = r_h^k + \mathbb{P}_h V_{h+1}^k - Q_h^k, \quad \forall (k,h) \in [K] \times [H]. \tag{4.1}$$

Here $V_{h+1}^k$ and $Q_h^k$ are the estimated value function and Q-function defined in (3.4), and $\mathbb{P}_h$ is the operator defined in (2.2). Intuitively, (4.1) quantifies the violation of the Bellman equation in (2.1). With these notations, we have the following regret decomposition lemma.

**Lemma 4.1** (Regret Decomposition). It holds that

$$
\begin{aligned}
\mathrm{Regret}(K) &= \sum_{k=1}^{K} \big(V_1^{\pi^*,k}(x_1^k) - V_1^{\pi^k,k}(x_1^k)\big) \\
&= \underbrace{\sum_{k=1}^{K} \sum_{h=1}^{H} \mathbb{E}_{\pi^*}\big[\langle Q_h^k(x_h,\cdot), \pi_h^*(\cdot \,|\, x_h) - \pi_h^k(\cdot \,|\, x_h)\rangle\big]}_{\text{policy optimization error}} \\
&\quad + \underbrace{\sum_{k=1}^{K} \sum_{h=1}^{H} \big(\mathbb{E}_{\pi^*}[\delta_h^k(x_h,a_h)] - \mathbb{E}_{\pi^k}[\delta_h^k(x_h,a_h)]\big)}_{\text{statistical error}}.
\end{aligned}
$$

*Proof.* This lemma is similar to the regret decomposition lemma in previous works [6, 30] on policy optimization. See §B.1 for a detailed proof. □

Lemma 4.1 shows that the total regret consists of the *policy optimization error* and the *statistical error* related to the Bellman error defined in (4.1). Notably, different from previous works [6, 30, 37, 14] that optimize policy in each episode, our algorithm performs policy optimization infrequently. Despite this, we can bound the policy optimization error in Lemma 4.1 by the following lemma.

**Lemma 4.2.** It holds that

$$\sum_{k=1}^{K} \sum_{h=1}^{H} \mathbb{E}_{\pi^*} \big[ \langle Q_h^k(x_h, \cdot), \pi_h^*(\cdot \,|\, x_h) - \pi_h^k(\cdot \,|\, x_h) \rangle \big] \leq \sqrt{2BH^4 K \cdot \log |\mathcal{A}|}.$$

*Proof.* See §B.2 for a detailed proof. □

For the statistical error in Lemma 4.1, by the definitions of the policy evaluation step in (3.4) and Bellman error in (4.1), we have

$$\delta_h^k = \underbrace{r_h^k - \bar{r}_h^k}_{\text{reward mismatch error}} + \underbrace{\mathbb{P}_h V_{h+1}^k - \widehat{\mathbb{P}}_h^k V_{h+1}^k}_{\text{transition estimation error}}. \tag{4.2}$$

The following lemma establishes the upper bound of the cumulative reward mismatch error

$$\sum_{k=1}^{K} \sum_{h=1}^{H} \big( \mathbb{E}_{\pi^*}[(r_h^k - \bar{r}_h^k)(x_h, a_h)] - \mathbb{E}_{\pi^k}[(r_h^k - \bar{r}_h^k)(x_h, a_h)] \big), \tag{4.3}$$

and thus relates the statistical error in Lemma 4.1 to the transition estimation error in (4.2).

**Lemma 4.3.** It holds with probability at least $1 - \delta/2$ that

$$\sum_{k=1}^{K} \sum_{h=1}^{H} \big( \mathbb{E}_{\pi^*}[\delta_h^k(x_h, a_h)] - \mathbb{E}_{\pi^k}[\delta_h^k(x_h, a_h)] \big)$$

$$\leq \sum_{k=1}^{K} \sum_{h=1}^{H} \mathbb{E}_{\pi^*}[(\mathbb{P}_h V_{h+1}^{t_k} - \widehat{\mathbb{P}}_h^{t_k} V_{h+1}^{t_k})(x_h, a_h)] + \sum_{k=1}^{K} \sum_{h=1}^{H} (\widehat{\mathbb{P}}_h^{t_k} V_{h+1}^{t_k} - \mathbb{P}_h V_{h+1}^{t_k})(x_h^k, a_h^k)$$

$$+ BH + \alpha H^3 K + \sqrt{H^3 K \iota}.$$

*Proof.* By calculation, we can show that the cumulative reward mismatch error in (4.3) is bounded by

$$\sum_{i=1}^{L-1} \sum_{k=k_i}^{k_{i+1}-1} \big( V_1^{\pi^{k_{i+1}}, k}(x_1) - V_1^{\pi^{k_i}, k}(x_1) \big) + \text{error terms},$$

which represents the smoothness of adjacent policies. Our smoothness analysis leverages the value difference lemma (Lemma B.1 or §B.1 in Cai et al. [6]) and the closed form of the policy improvement in (3.2). See §B.3 for a detailed proof. □

Then we introduce the following lemma, which shows that the transition estimation error can be controlled by the bonus function.

**Lemma 4.4.** It holds with probability at least $1 - \delta/2$ that

$$-2 \min\{H, \Gamma_h^{t_k}(x, a)\} \leq (\mathbb{P}_h V_{h+1}^{t_k} - \widehat{\mathbb{P}}_h^{t_k} V_{h+1}^{t_k})(x, a) \leq 0$$

for all $(k, h, x, a) \in [K] \times [H] \times \mathcal{S} \times \mathcal{A}$.

*Proof.* The proof involves the standard analysis of self-normalized process [1] and a uniform concentration of the function class of $V_{h+1}^k$ [17]. As elaborated in §1.1, calculating the covering number of this function class is challenging and requires some new techniques. See §B.4 for a detailed proof. □

Combining Lemmas 4.3 and 4.4, we have

$$\sum_{k=1}^{K}\sum_{h=1}^{H}\left(\mathbb{E}_{\pi^*}[\delta_h^k(x_h,a_h)] - \mathbb{E}_{\pi^k}[\delta_h^k(x_h,a_h)]\right)$$

$$\leq 2\sum_{k=1}^{K}\sum_{h=1}^{H}\min\{H,\Gamma_h^{t_k}(x_h^k,a_h^k)\} + BH + \alpha H^3 K + \sqrt{H^3 K\iota}.$$

Hence, it remains to bound the term $\sum_{k=1}^{K}\sum_{h=1}^{H}\min\{H,\Gamma_h^{t_k}(x_h^k,a_h^k)\}$, which is the purpose of the following lemma.

**Lemma 4.5.** It holds that

$$\sum_{k=1}^{K}\sum_{h=1}^{H}\min\{H,\Gamma_h^{t_k}(x_h^k,a_h^k)\} \leq \mathcal{O}(d^{3/4}H^2K^{3/4}\cdot\iota) + \mathcal{O}(d^{5/2}H^2K^{1/2}\cdot\iota).$$

*Proof.* For all $k\in[K]$, let $\Gamma_h^k = \beta\cdot(\phi^\top(\Lambda_h^k)^{-1}\phi)^{1/2}$ with $\Lambda_h^k = \sum_{\tau=1}^{k-1}\phi(x_h^\tau,a_h^\tau)\phi(x_h^\tau,a_h^\tau)^\top + \lambda\cdot I_d$. By a doubling trick, we can prove that

$$\sum_{k=1}^{K}\sum_{h=1}^{H}\min\{H,\Gamma_h^{t_k}(x_h^k,a_h^k)\} \lesssim \sum_{k=1}^{K}\sum_{h=1}^{H}\min\{H,\Gamma_h^k(x_h^k,a_h^k)\} + \text{error terms},$$

which can be further bounded by elliptical potential lemma (Lemma D.5 or Lemma 11 in Abbasi-Yadkori et al. [1]). See §B.5 for a detailed proof. □

Finally, putting Lemmas 4.1-4.5 together, we conclude the proof of Theorem 3.1. □

Note the choice of the number of batches $L$ determines the batch size $B = K/L$. We make the following remark to illustrate how the choice of $L$ affects the final regret, which indicates that our choices of parameters are optimal based on our current analysis.

**Remark 4.6** (Choices of Parameters and Final Regret). If OPPO+ performs the policy optimization for $L$ times, then by Lemma C.3, we have the complexity (logarithmic covering number) of policy class is roughly $\widetilde{\mathcal{O}}(L)$ (ignoring the dependency of $d$ and $H$). Furthermore, by the new self-normalized analysis in §C, we have $\beta = \widetilde{\mathcal{O}}(\sqrt{L})$, which further implies that the model estimation error is bounded by $\beta\cdot\sum_{k=1}^{K}\sum_{h=1}^{H}\sqrt{\phi(x_h^k,a_h^k)(\Lambda_h^k)^{-1}\phi(x_h^k,a_h^k)} \leq \widetilde{\mathcal{O}}(\sqrt{LK})$ (cf. Lemma 4.5). Meanwhile, by Lemma 4.2, we have the policy optimization error is bounded by $\widetilde{\mathcal{O}}(K/\sqrt{L}) = \widetilde{\mathcal{O}}(K^{3/4})$. By choosing $L = \Theta(\sqrt{K})$, we obtain a regret of order $\widetilde{\mathcal{O}}(K^{3/4})$.

## 5 Conclusion

In this paper, we advance the theoretical study of PPO in stochastic linear MDPs and even adversarial linear MDPs with full-information feedback. We propose a novel algorithm, namely OPPO+, which exhibits a state-of-the-art regret bound as compared to the prior policy optimization algorithms. Our work paves the way for future research in multiple directions. For instance, a significant open research question is to investigate whether policy-based algorithms can achieve the minimax regret in stochastic linear MDPs like previous value-based algorithms [3, 14]. Additionally, an interesting direction is to derive $\sqrt{K}$-regret bounds for adversarial linear MDPs with full-information or even bandit feedback.

## Acknowledgements

This work was done while Han Zhong visiting HKUST. The authors would like to thank Miao Lu, Haipeng Luo, Tianhao Wu, and Wei Xiong for helpful discussions and feedback.

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

# A   Further Elaboration of Challenge 1 in Section 1.1

Regarding the technical challenges involved in extending OPPO [6] to linear MDPs, we provided more detailed explanations below.

Technically, for linear mixture MDPs (Equation (B.20) in OPPO paper [6]), they need to analyze

$$\left\| \sum_{\tau=1}^{k-1} \phi_h^\tau(x_h^\tau, a_h^\tau) \cdot \left( V_{h+1}^\tau(x_{h+1}^\tau) - (\mathbb{P}_h V_{h+1}^\tau)(x_h^\tau, a_h^\tau) \right) \right\|_{(\Lambda_h^k)^{-1}}.$$

Since $V_{h+1}^\tau$ is adapted to $\mathcal{F}_{k,h,1} = \{(x_i^\tau, a_i^\tau)\}_{(\tau,i)\in[k-1]\times[H]} \cup \{r^\tau\}_{\tau\in[k]} \cup \{(x_i^k, a_i^k)\}_{i\in[h]}$, they can bound this term with classical self-normalized process analysis directly (see Lemma D.1 in Cai et al. [6]).

In contrast, for linear MDPs (see e.g., (B.26) in our paper or Lemma B.3 in Jin et al. [17]), we need to bound the term

$$\left\| \sum_{\tau=1}^{k-1} \phi(x_h^\tau, a_h^\tau) \cdot \left( V_{h+1}^k(x_{h+1}^\tau) - (\mathbb{P}_h V_{h+1}^k)(x_h^\tau, a_h^\tau) \right) \right\|_{(\Lambda_h^k)^{-1}}.$$

Since $V_{h+1}^\tau$ is NOT adapted to $\mathcal{F}_{k,h,1} = \{(x_i^\tau, a_i^\tau)\}_{(\tau,i)\in[k-1]\times[H]} \cup \{r^\tau\}_{\tau\in[k]} \cup \{(x_i^k, a_i^k)\}_{i\in[h]}$. We need to perform the uniform concentration on the function class of $V_{h+1}^k$. The challenge of calculating the covering number of this function class has been elaborated in Challenge 1 of Section 1.1.

# B   Missing Proofs of Main Theorem

## B.1   Proof of Lemma 4.1

*Proof.* Our proof relies on the following value difference lemma in Cai et al. [6].

**Lemma B.1** (Value Difference Lemma). Let $\pi = \{\pi_h\}_{h\in[H]}$ and $\pi' = \{\pi_h'\}_{h\in[H]}$ be two policies and $\bar{Q} = \{\bar{Q}_h : \mathcal{S} \times \mathcal{A} \mapsto \mathbb{R}\}_{h\in[H]}$ be any Q-functions. Moreover, for any $h \in [H]$, we define value function $\bar{V}_h : \mathcal{S} \mapsto \mathbb{R}$ by letting $\bar{V}_h(x) = \langle \bar{Q}_h(x,\cdot), \pi_h(\cdot \mid x) \rangle$. Then for any $k \in [K]$ we have

$$\bar{V}_1(x_1) - V_1^{\pi',k}(x_1) = \sum_{h=1}^H \mathbb{E}_{\pi'}[\langle \bar{Q}_h(x_h,\cdot), \pi_h(\cdot \mid x_h) - \pi_h'(\cdot \mid x_h) \rangle]$$

$$+ \sum_{h=1}^H \mathbb{E}_{\pi'}[\bar{Q}_h(x_h, a_h) - (r_h^k + \mathbb{P}_h \bar{V}_{h+1})(x_h, a_h)].$$

*Proof.* See §B.1 in Cai et al. [6] for a detailed proof. $\square$

Back to our proof, for any $k \in [K]$, we have

$$V_1^{\pi^*,k}(x_1) - V_1^{\pi^k,k}(x_1) = \underbrace{V_1^{\pi^*,k}(x_1) - V_1^k(x_1)}_{\text{(i)}} + \underbrace{V_1^k(x_1) - V_1^{\pi^k,k}(x_1)}_{\text{(ii)}}. \quad (B.1)$$

Applying Lemma B.1 with $\pi = \pi^k$, $\pi' = \pi^*$, and $\bar{Q} = \{Q_h^k\}_{h\in[H]}$, we have

$$\text{(i)} = \sum_{k=1}^K \sum_{h=1}^H \mathbb{E}_{\pi^*}\left[\langle Q_h^k(x_h,\cdot), \pi_h^*(\cdot \mid x_h) - \pi_h^k(\cdot \mid x_h)\rangle\right] + \sum_{k=1}^K \sum_{h=1}^H \mathbb{E}_{\pi^*}[\delta_h^k(x_h, a_h)], \quad (B.2)$$

where $\delta_h^k = r_h^k + Q_h^k - \mathbb{P}_h V_{h+1}^k$ is the Bellman error defined in (4.1). Similarly, applying Lemma B.1 with $\pi = \pi' = \pi^k$ and $\bar{Q} = \{Q_h^k\}_{h\in[H]}$, we obtain

$$\text{(ii)} = -\sum_{k=1}^K \sum_{h=1}^H \mathbb{E}_{\pi^k}[\delta_h^k(x_h, a_h)]. \quad (B.3)$$

Plugging (B.2) and (B.3) into (B.1) and then taking summation across $k \in [K]$, we have

$$
\begin{aligned}
\text{Regret}(K) &= \sum_{k=1}^{K} \left( V_1^{\pi^*,k}(x_1^k) - V_1^{\pi^k,k}(x_1^k) \right) \\
&= \sum_{k=1}^{K} \sum_{h=1}^{H} \mathbb{E}_{\pi^*} \left[ \langle Q_h^k(x_h, \cdot), \pi_h^*(\cdot \mid x_h) - \pi_h^k(\cdot \mid x_h) \rangle \right] \\
&\quad + \sum_{k=1}^{K} \sum_{h=1}^{H} \left( \mathbb{E}_{\pi^*}[\delta_h^k(x_h, a_h)] - \mathbb{E}_{\pi^k}[\delta_h^k(x_h, a_h)] \right),
\end{aligned}
$$

which concludes the proof of Lemma 4.1. $\qquad \square$

## B.2 Proof of Lemma 4.2

*Proof.* Recall that $B \in \mathbb{N}^+$ is the batch size, $k_i = (i-1) \cdot B + 1$ and $L = K/B$. Fix $h \in [H]$. By the multi-batched updating rule, we have

$$
\sum_{k=1}^{K} \mathbb{E}_{\pi^*} \left[ \langle Q_h^k(x_h, \cdot), \pi_h^*(\cdot \mid x_h) - \pi_h^k(\cdot \mid x_h) \rangle \right] = B \sum_{i=1}^{L} \mathbb{E}_{\pi^*} \left[ \langle Q_h^{k_i}(x_h, \cdot), \pi_h^*(\cdot \mid x_h) - \pi_h^{k_i}(\cdot \mid x_h) \rangle \right].
\tag{B.4}
$$

To derive the upper bound of (B.4), we need the following lemma.

**Lemma B.2.** For any $(i, h, x_h) \in [L] \times [H] \times \mathcal{S}$, it holds that

$$
\begin{aligned}
&\langle Q_h^{k_i}(x_h, \cdot), \pi_h^*(\cdot \mid x_h) - \pi_h^{k_i}(\cdot \mid x_h) \rangle \\
&\qquad \leq \frac{\alpha H^2}{2} + \frac{\text{KL}\left(\pi_h^*(\cdot \mid x_h) \,\big\|\, \pi_h^{k_i}(\cdot \mid x_h)\right) - \text{KL}\left(\pi_h^*(\cdot \mid x_h) \,\big\|\, \pi_h^{k_{i+1}}(\cdot \mid x_h)\right)}{\alpha}.
\end{aligned}
$$

*Proof.* By the updating rule in (3.2), we have

$$
\pi_h^{k_{i+1}}(\cdot \mid x_h) = \frac{\pi_h^{k_i}(\cdot \mid x_h) \cdot \exp\{\alpha Q_h^{k_i}(x_h, \cdot)\}}{\sum_{a \in \mathcal{A}} \pi_h^{k_i}(a \mid x_h) \cdot \exp\{\alpha Q_h^{k_i}(x_h, a)\}}.
\tag{B.5}
$$

For ease of presentation, we denote $\Upsilon = \sum_{a \in \mathcal{A}} \pi_h^{k_i}(a \mid x_h) \cdot \exp\{\alpha Q_h^{k_i}(x_h, a)\}$. Then we have

$$
\begin{aligned}
&\langle \alpha Q_h^{k_i}(x_h, \cdot), \pi_h^*(\cdot \mid x_h) - \pi_h^{k_{i+1}}(\cdot \mid x_h) \rangle \\
&\quad = \langle \log \Upsilon + \log \pi_h^{k_{i+1}}(\cdot \mid x_h) - \log \pi_h^{k_i}(\cdot \mid x_h), \pi_h^*(\cdot \mid x_h) - \pi_h^{k_{i+1}}(\cdot \mid x_h) \rangle \\
&\quad = \langle \log \pi_h^{k_{i+1}}(\cdot \mid x_h) - \log \pi_h^{k_i}(\cdot \mid x_h), \pi_h^*(\cdot \mid x_h) - \pi_h^{k_{i+1}}(\cdot \mid x_h) \rangle
\end{aligned}
\tag{B.6}
$$

where the first equality uses (B.5), and the second equality follows from the fact that $\sum_{a \in \mathcal{A}} \pi_h^*(a \mid x_h) - \pi_h^{k_{i+1}}(a \mid x_h) = 0$. Rearranging (B.6) gives that

$$
\begin{aligned}
(\text{B.6}) &= \left\langle \log\left(\frac{\pi_h^*(\cdot \mid x_h)}{\pi_h^{k_i}(\cdot \mid x_h)}\right), \pi_h^*(\cdot \mid x_h) \right\rangle - \left\langle \log\left(\frac{\pi_h^*(\cdot \mid x_h)}{\pi_h^{k_{i+1}}(\cdot \mid x_h)}\right), \pi_h^*(\cdot \mid x_h) \right\rangle \\
&\quad - \left\langle \log\left(\frac{\pi_h^{k_{i+1}}(\cdot \mid x_h)}{\pi_h^{k_i}(\cdot \mid x_h)}\right), \pi_h^{k_{i+1}}(\cdot \mid x_h) \right\rangle \\
&= \text{KL}\left(\pi_h^*(\cdot \mid x_h) \,\big\|\, \pi_h^{k_i}(\cdot \mid x_h)\right) - \text{KL}\left(\pi_h^*(\cdot \mid x_h) \,\big\|\, \pi_h^{k_{i+1}}(\cdot \mid x_h)\right) \\
&\quad - \text{KL}\left(\pi_h^{k_{i+1}}(\cdot \mid x_h) \,\big\|\, \pi_h^{k_i}(\cdot \mid x_h)\right).
\end{aligned}
\tag{B.7}
$$

Furthermore, we have

$$
\begin{aligned}
&\langle \alpha Q_h^{k_i}(x_h, \cdot), \pi_h^*(\cdot \mid x_h) - \pi_h^{k_i}(\cdot \mid x_h) \rangle \\
&\quad = \langle \alpha Q_h^{k_i}(x_h, \cdot), \pi_h^*(\cdot \mid x_h) - \pi_h^{k_{i+1}}(\cdot \mid x_h) \rangle - \langle \alpha Q_h^{k_i}(x_h, \cdot), \pi_h^{k_i}(\cdot \mid x_h) - \pi_h^{k_{i+1}}(\cdot \mid x_h) \rangle \\
&\quad \leq \text{KL}\left(\pi_h^*(\cdot \mid x_h) \,\big\|\, \pi_h^{k_i}(\cdot \mid x_h)\right) - \text{KL}\left(\pi_h^*(\cdot \mid x_h) \,\big\|\, \pi_h^{k_{i+1}}(\cdot \mid x_h)\right) \\
&\quad\quad - \text{KL}\left(\pi_h^{k_{i+1}}(\cdot \mid x_h) \,\big\|\, \pi_h^{k_i}(\cdot \mid x_h)\right) + \alpha H \cdot \|\pi_h^{k_i}(\cdot \mid x_h) - \pi_h^{k_{i+1}}(\cdot \mid x_h)\|_1,
\end{aligned}
\tag{B.8}
$$

where the last inequality uses (B.7), Cauchy-Schwarz inequality, and the fact that $\|Q_h^{k_i}\|_\infty \leq H$. By Pinsker's inequality, we have $\mathrm{KL}(\pi_h^{k_{i+1}}(\cdot \mid x_h) \,\|\, \pi_h^{k_i}(\cdot \mid x_h)) \geq \|\pi_h^{k_{i+1}}(\cdot \mid x_h) - \pi_h^{k_i}(\cdot \mid x_h)\|_1^2/2$. Together with (B.8), we obtain that

$$
\langle \alpha Q_h^{k_i}(x_h, \cdot), \pi_h^*(\cdot \mid x_h) - \pi_h^{k_i}(\cdot \mid x_h) \rangle
$$
$$
\leq \mathrm{KL}\big(\pi_h^*(\cdot \mid x_h) \,\|\, \pi_h^{k_i}(\cdot \mid x_h)\big) - \mathrm{KL}\big(\pi_h^*(\cdot \mid x_h) \,\|\, \pi_h^{k_{i+1}}(\cdot \mid x_h)\big)
$$
$$
- \|\pi_h^{k_{i+1}}(\cdot \mid x_h) - \pi_h^{k_i}(\cdot \mid x_h)\|_1^2/2 + \alpha H \cdot \|\pi_h^{k_i}(\cdot \mid x_h) - \pi_h^{k_{i+1}}(\cdot \mid x_h)\|_1
$$
$$
\leq \mathrm{KL}\big(\pi_h^*(\cdot \mid x_h) \,\|\, \pi_h^{k_i}(\cdot \mid x_h)\big) - \mathrm{KL}\big(\pi_h^*(\cdot \mid x_h) \,\|\, \pi_h^{k_{i+1}}(\cdot \mid x_h)\big) + \alpha^2 H^2/2, \qquad \text{(B.9)}
$$

where the last inequality uses the fact that $\max_{y \in \mathbb{R}}\{-y^2/2 + \alpha H \cdot y\} = \alpha^2 H^2/2$. Rearranging (B.9) concludes the proof of Lemma B.2. $\qquad \square$

By Lemma B.2, we further have

$$
\text{(B.4)} \leq B \sum_{i=1}^{L} \Big( \frac{\alpha H^2}{2} + \frac{\mathbb{E}_{\pi^*}\big[ \mathrm{KL}\big(\pi_h^*(\cdot \mid x_h) \,\|\, \pi_h^{k_i}(\cdot \mid x_h)\big) - \mathrm{KL}\big(\pi_h^*(\cdot \mid x_h) \,\|\, \pi_h^{k_{i+1}}(\cdot \mid x_h)\big) \big]}{\alpha} \Big)
$$
$$
= B \cdot \Big( \frac{\alpha H^2 K}{2B} + \frac{\mathbb{E}_{\pi^*}\big[ \mathrm{KL}\big(\pi_h^*(\cdot \mid x_h) \,\|\, \pi_h^{k_1}(\cdot \mid x_h)\big) - \mathrm{KL}\big(\pi_h^*(\cdot \mid x_h) \,\|\, \pi_h^{k_{L+1}}(\cdot \mid x_h)\big) \big]}{\alpha} \Big)
$$
$$
\leq \frac{\alpha H^2 K}{2} + \frac{B \cdot \log|\mathcal{A}|}{\alpha}, \qquad\qquad\qquad\qquad\qquad\qquad\qquad\qquad\qquad\qquad \text{(B.10)}
$$

where the equality uses the fact that $L = K/B$, and the last inequality uses the non-negativity of KL-divergence and the fact that $\pi_h^{k_1} = \pi_h^1$ is the uniform policy. Combining (B.4), (B.10), and $\alpha = \sqrt{2B \log|\mathcal{A}|/(KH^2)}$, we obtain for all $h \in [H]$:

$$
\sum_{k=1}^{K} \mathbb{E}_{\pi^*}\big[\langle Q_h^k(x_h, \cdot), \pi_h^*(\cdot \mid x_h) - \pi_h^k(\cdot \mid x_h)\rangle\big] \leq \sqrt{2BH^2 K \cdot \log|\mathcal{A}|}. \qquad \text{(B.11)}
$$

Telescoping (B.11) across $h \in [H]$ concludes the proof of Lemma 4.2. $\qquad \square$

## B.3 Proof of Lemma 4.3

*Proof.* Recall that $B \in \mathbb{N}^+$ is the batch size, $k_i = (i-1) \cdot B + 1$ and $L = K/B$. Fix $h \in [H]$. We have

$$
\sum_{k=1}^{K} \mathbb{E}_{\pi^*}[\delta_h^k(x_h, a_h)] = \sum_{k=1}^{K} \mathbb{E}_{\pi^*}[r_h^k(s_h, a_h) + \mathbb{P}_h V_{h+1}^k(x_h, a_h) - Q_h^k(x_h, a_h)]
$$
$$
= \sum_{k=1}^{K} \mathbb{E}_{\pi^*}[r_h^k(s_h, a_h) + \mathbb{P}_h V_{h+1}^{t_k}(x_h, a_h) - Q_h^{t_k}(x_h, a_h)] \qquad \text{(B.12)}
$$
$$
= \sum_{k=1}^{K} \mathbb{E}_{\pi^*}[r_h^k(s_h, a_h) - \bar{r}_h^{t_k}(x_h, a_h)] + \sum_{k=1}^{K} \mathbb{E}_{\pi^*}[(\mathbb{P}_h V_{h+1}^{t_k} - \widehat{\mathbb{P}}_h^{t_k} V_{h+1}^{t_k})(x_h, a_h)],
$$

where the first equality uses the definition of $t_k$ and the updating rule, and the second equality uses the definition of $Q_h^k = \bar{r}_h^k + \widehat{\mathbb{P}}_h^k V_{h+1}^k$ in (3.4). Furthermore, we have

$$
\sum_{k=1}^{K} \mathbb{E}_{\pi^*}[r_h^k(x_h, a_h) - \bar{r}_h^{t_k}(x_h, a_h)] = \sum_{k=1}^{K} \mathbb{E}_{\pi^*}[r_h^k(x_h, a_h)] - B \sum_{i=1}^{L} \mathbb{E}_{\pi^*}[\bar{r}_h^{k_i}(x_h, a_h)]
$$
$$
= \sum_{k=1}^{K} \mathbb{E}_{\pi^*}[r_h^k(x_h, a_h)] - \sum_{i=1}^{L} \sum_{k=k_{i-1}}^{k_i - 1} \mathbb{E}_{\pi^*}[r_h^k(x_h, a_h)]
$$
$$
= \sum_{k=K-B+1}^{K} \mathbb{E}_{\pi^*}[r_h^k(x_h, a_h)] \leq B, \qquad \text{(B.13)}
$$

where the first equality uses the updating rule, the second equality follows from the definition of $\bar{r}_h^k$ in (3.5), and the last inequality is obtained by the fact that $\|r_h^k\|_\infty \le 1$ for any $(k, h) \in [K] \times [H]$. Combining (B.12) and (B.13) and then taking summation across $h \in [H]$, we obtain that

$$\sum_{k=1}^K \sum_{h=1}^H \mathbb{E}_{\pi^*}[\delta_h^k(x_h, a_h)] \le \sum_{k=1}^K \sum_{h=1}^H \mathbb{E}_{\pi^*}[(\mathbb{P}_h V_{h+1}^{t_k} - \widehat{\mathbb{P}}_h^{t_k} V_{h+1}^{t_k})(x_h, a_h)] + BH. \qquad (B.14)$$

On the other hand, similar to the derivation of (B.12), we have

$$-\sum_{k=1}^K \mathbb{E}_{\pi^k}[\delta_h^k(x_h, a_h)]$$

$$= \sum_{k=1}^K \mathbb{E}_{\pi^{t_k}}[\bar{r}_h^{t_k}(x_h, a_h) - r_h^k(s_h, a_h)] + \sum_{k=1}^K \mathbb{E}_{\pi^k}[(\widehat{\mathbb{P}}_h^{t_k} V_{h+1}^{t_k} - \mathbb{P}_h V_{h+1}^{t_k})(x_h, a_h)]. \qquad (B.15)$$

For the first term of (B.15), by the updating rule and calculation, we have

$$\sum_{k=1}^K \mathbb{E}_{\pi^{t_k}}[\bar{r}_h^{t_k}(x_h, a_h) - r_h^k(s_h, a_h)] = B \sum_{i=1}^L \mathbb{E}_{\pi^{k_i}}[\bar{r}_h^{k_i}(x_h, a_h)] - \sum_{i=1}^L \sum_{k=k_i}^{k_{i+1}-1} \mathbb{E}_{\pi^{k_i}}[r_h^k(x_h, a_h)]$$

$$= \sum_{i=1}^L \sum_{k=k_{i-1}}^{k_i-1} \mathbb{E}_{\pi^{k_i}}[r_h^k(x_h, a_h)] - \sum_{i=1}^L \sum_{k=k_i}^{k_{i+1}-1} \mathbb{E}_{\pi^{k_i}}[r_h^k(x_h, a_h)]$$

$$\le \sum_{i=1}^{L-1} \sum_{k=k_i}^{k_{i+1}-1} \left(\mathbb{E}_{\pi^{k_{i+1}}}[r_h^k(x_h, a_h)] - \mathbb{E}_{\pi^{k_i}}[r_h^k(x_h, a_h)]\right),$$

$$(B.16)$$

where the last inequality uses the fact that $-\sum_{k=k_L}^{k_{L+1}-1} \mathbb{E}_{\pi^{k_L}}[r_h^k(x_h, a_h)] \le 0$. Summing over $h \in [H]$ in (B.16) gives that

$$\sum_{k=1}^K \sum_{h=1}^H \mathbb{E}_{\pi^{t_k}}[\bar{r}_h^{t_k}(x_h, a_h) - r_h^k(s_h, a_h)]$$

$$\le \sum_{i=1}^{L-1} \sum_{k=k_i}^{k_{i+1}-1} \left(V_1^{\pi^{k_{i+1}}, k}(x_1) - V_1^{\pi^{k_i}, k}(x_1)\right)$$

$$= \sum_{i=1}^{L-1} \sum_{k=k_i}^{k_{i+1}-1} \sum_{h=1}^H \mathbb{E}_{\pi^{k_{i+1}}}[\langle Q_h^{\pi^{k_i}, k}(\cdot \mid x_h), \pi_h^{k_{i+1}}(\cdot \mid x_h) - \pi_h^{k_i}(\cdot \mid x_h)\rangle], \qquad (B.17)$$

where the last inequality uses the value difference lemma (Lemma B.1). By the policy updating rule in (3.2), we have

$$\pi_h^{k_{i+1}}(\cdot \mid x_h) = \frac{\pi_h^{k_i}(\cdot \mid x_h) \cdot \exp\left(\alpha Q_h^{k_i}(x_h, \cdot)\right)}{\sum_{a \in \mathcal{A}} \pi_h^{k_i}(a \mid x_h) \cdot \exp\left(\alpha Q_h^{k_i}(x_h, a)\right)}.$$

for any $x_h \in \mathcal{S}$, which implies that

$$\frac{\pi_h^{k_i}(\cdot \mid x_h)}{\pi_h^{k_{i+1}}(\cdot \mid x_h)} = \frac{\sum_{a \in \mathcal{A}} \pi_h^{k_i}(a \mid x_h) \cdot \exp\left(\alpha Q_h^{k_i}(x_h, a)\right)}{\exp\left(\alpha Q_h^{k_i}(x_h, \cdot)\right)}$$

$$\ge \frac{\sum_{a \in \mathcal{A}} \pi_h^{k_i}(a \mid x_h)}{\exp(\alpha H)} = \exp(-\alpha H) \ge 1 - \alpha H,$$

where the first inequality follows the fact that $0 \le Q_h^{k_i}(\cdot, \cdot) \le H$, and the last inequality uses the basic inequality $\exp(y) \ge 1 + y$ for all $y \in \mathbb{R}$. Together with

$$\pi_h^{k_{i+1}}(\cdot \mid x_h) - \pi_h^{k_i}(\cdot \mid x_h) = \pi_h^{k_{i+1}}(\cdot \mid x_h) \cdot \left(1 - \frac{\pi_h^{k_i}(\cdot \mid x_h)}{\pi_h^{k_{i+1}}(\cdot \mid x_h)}\right),$$

we further obtain

$$\pi_h^{k_{i+1}}(\cdot \mid x_h) - \pi_h^{k_i}(\cdot \mid x_h) \le \alpha H \cdot \pi_h^{k_{i+1}}(\cdot \mid x_h) \tag{B.18}$$

for any $x_h \in \mathcal{S}$. Plugging (B.18) into (B.17) gives that

$$(\text{B.17}) \le \alpha H \sum_{i=1}^{L-1} \sum_{k=k_i}^{k_{i+1}-1} \sum_{h=1}^{H} \mathbb{E}_{\pi^{k_{i+1}}} [\langle Q_h^{\pi^{k_i},k}(x_h, \cdot), \pi_h^{k_{i+1}}(\cdot \mid x_h) \rangle] \le \alpha H^3 K, \tag{B.19}$$

where the last inequality follows from the fact that $Q_h^{\pi^{k_i},k}(\cdot, \cdot) \le H$. Plugging (B.17) and (B.19) into (B.15), we have

$$-\sum_{k=1}^{K} \sum_{h=1}^{H} \mathbb{E}_{\pi^k}[\delta_h^k(x_h, a_h)] \le \sum_{k=1}^{K} \sum_{h=1}^{H} \mathbb{E}_{\pi^k}[(\widehat{\mathbb{P}}_h^{t_k} V_{h+1}^{t_k} - \mathbb{P}_h V_{h+1}^{t_k})(x_h, a_h)] + \alpha H^3 K$$

$$\le \sum_{k=1}^{K} \sum_{h=1}^{H} (\widehat{\mathbb{P}}_h^{t_k} V_{h+1}^{t_k} - \mathbb{P}_h V_{h+1}^{t_k})(x_h^k, a_h^k) + \alpha H^3 K + \sqrt{H^3 K \iota}, \tag{B.20}$$

where the last inequality uses Azuma-Hoeffding inequality. Putting (B.14) and (B.20) together, we conclude the proof of Lemma 4.3. $\qquad\square$

## B.4   Proof of Lemma 4.4

*Proof.* Recall that

$$(\widehat{\mathbb{P}}_h^{t_k} V_{h+1}^{t_k})(\cdot, \cdot) = \min\{\phi(\cdot, \cdot)^\top w_h^{t_k} + \Gamma_h^{t_k}(\cdot, \cdot), H - h\}^+, \tag{B.21}$$

where $w_h^{t_k}$ and $\Gamma_h^{t_k}$ take form

$$w_h^{t_k} = (\Lambda_h^{t_k})^{-1} \Big( \sum_{\tau=1}^{t_k-1} \phi(x_h^\tau, a_h^\tau) \cdot V_{h+1}^{t_k}(x_{h+1}^\tau) \Big), \quad \Gamma_h^{t_k}(\cdot, \cdot) = \beta \cdot \sqrt{\phi(\cdot, \cdot)^\top (\Lambda_h^{t_k})^{-1} \phi(\cdot, \cdot)}. \tag{B.22}$$

Here $\Lambda_h^{t_k}$ is the covariance matrix:

$$\Lambda_h^{t_k} = \sum_{\tau=1}^{t_k-1} \phi(x_h^\tau, a_h^\tau) \phi(x_h^\tau, a_h^\tau)^\top + \lambda \cdot I_d. \tag{B.23}$$

Back to our proof, for any $(x, a) \in \mathcal{S} \times \mathcal{A}$, we have

$$(\mathbb{P}_h V_{h+1}^{t_k})(x, a) = \phi(x, a)^\top \langle \mu_h, V_{h+1}^{t_k} \rangle_{\mathcal{S}}$$

$$= \phi(x, a)^\top (\Lambda_h^{t_k})^{-1} \Lambda_h^{t_k} \langle \mu_h, V_{h+1}^{t_k} \rangle_{\mathcal{S}}$$

$$= \phi(x, a)^\top (\Lambda_h^{t_k})^{-1} \Big( \sum_{\tau=1}^{t_k-1} \phi(x_h^\tau, a_h^\tau) \phi(x_h^\tau, a_h^\tau)^\top \langle \mu_h, V_{h+1}^{t_k} \rangle_{\mathcal{S}} + \lambda \cdot \langle \mu_h, V_{h+1}^{t_k} \rangle_{\mathcal{S}} \Big)$$

$$= \phi(x, a)^\top (\Lambda_h^{t_k})^{-1} \Big( \sum_{\tau=1}^{t_k-1} \phi(x_h^\tau, a_h^\tau) \cdot (\mathbb{P}_h V_{h+1}^{t_k})(x_h^\tau, a_h^\tau) + \lambda \cdot \langle \mu_h, V_{h+1}^{t_k} \rangle_{\mathcal{S}} \Big), \tag{B.24}$$

where the first and the last equality follows from the definition of linear MDP (Definition 2.1), and the third equality uses (B.23). Putting (B.22) and (B.24) together, we have

$$\phi(x, a)^\top w_h^{t_k} - \mathbb{P}_h V_{h+1}^{t_k}(x, a)$$

$$= \underbrace{\phi(x, a)^\top (\Lambda_h^{t_k})^{-1} \Big( \sum_{\tau=1}^{t_k-1} \phi(x_h^\tau, a_h^\tau) \cdot \big( V_{h+1}^{t_k}(x_{h+1}^\tau) - (\mathbb{P}_h V_{h+1}^{t_k})(x_h^\tau, a_h^\tau) \big) \Big)}_{(\star)}$$

$$\underbrace{- \lambda \cdot \phi(x, a)^\top (\Lambda_h^k)^{-1} \langle \mu_h, V_{h+1}^{t_k} \rangle_{\mathcal{S}}}_{(\star\star)}. \tag{B.25}$$

For Term $(\star)$ in (B.25), by Cauchy-Schwarz inequality, we have

$$
(\star) \leq \sqrt{\phi(x,a)^\top (\Lambda_h^{t_k})^{-1} \phi(x,a)} \cdot \left\| \sum_{\tau=1}^{t_k-1} \phi(x_h^\tau, a_h^\tau) \cdot \left( V_{h+1}^{t_k}(x_{h+1}^\tau) - (\mathbb{P}_h V_{h+1}^{t_k})(x_h^\tau, a_h^\tau) \right) \right\|_{(\Lambda_h^{t_k})^{-1}}
$$

$$
\leq \mathcal{O}(d^{1/4} H K^{1/4} \iota^{1/2}) \cdot \sqrt{\phi(x,a)^\top (\Lambda_h^{t_k})^{-1} \phi(x,a)}, \tag{B.26}
$$

where the last inequality follows from the following lemma.

**Lemma B.3.** Fix $\delta \in (0,1]$. It holds for all $(k,h) \in [K] \times [H]$ that

$$
\left\| \sum_{\tau=1}^{t_k-1} \phi(x_h^\tau, a_h^\tau) \cdot \left( V_{h+1}^{t_k}(x_{h+1}^\tau) - (\mathbb{P}_h V_{h+1}^{t_k})(x_h^\tau, a_h^\tau) \right) \right\|_{(\Lambda_h^{t_k})^{-1}} \leq \mathcal{O}(d^{1/4} H K^{1/4} \iota^{1/2}).
$$

*Proof.* See §C for a detailed proof. $\qquad\square$

For Term $(\star\star)$ in (B.25), we have

$$
(\star\star) \leq \lambda \cdot \sqrt{\phi(x,a)^\top (\Lambda_h^k)^{-1} \phi(x,a)} \cdot \left\| \langle \mu_h, V_{h+1}^{t_k} \rangle_{\mathcal{S}} \right\|_{(\Lambda_h^{t_k})^{-1}}
$$

$$
\leq \sqrt{\lambda} \cdot \sqrt{\phi(x,a)^\top (\Lambda_h^k)^{-1} \phi(x,a)} \cdot \left\| \langle \mu_h, V_{h+1}^{t_k} \rangle_{\mathcal{S}} \right\|_2
$$

$$
\leq \sqrt{\lambda d H^2} \cdot \sqrt{\phi(x,a)^\top (\Lambda_h^k)^{-1} \phi(x,a)}, \tag{B.27}
$$

where the second inequality uses the fact that $\lambda \cdot I_d \preceq \Lambda_h^{t_k}$, and last inequality follows from $\|\langle \mu_h, V_{h+1}^{t_k} \rangle_{\mathcal{S}}\|_2 \leq H\sqrt{d}$, which is implied by Definition 2.1. Plugging (B.26) and (B.27) into (B.25), together with $\lambda = 1$, we obtain

$$
|\phi(x,a)^\top w_h^{t_k} - \mathbb{P}_h V_{h+1}^{t_k}(x,a)| \leq \beta \sqrt{\phi(x,a)^\top (\Lambda_h^{t_k})^{-1} \phi(x,a)} = \Gamma_h^{t_k}(x,a) \tag{B.28}
$$

for any $(x,a) \in \mathcal{S} \times \mathcal{A}$ with $\beta = \mathcal{O}(d^{1/4} H K^{1/4} \iota^{1/2})$. Together with the definition of $\widehat{\mathbb{P}}_h^{t_k} V_{h+1}^{t_k}$ in (B.21), we have

$$
(\mathbb{P}_h V_{h+1}^{t_k} - \widehat{\mathbb{P}}_h^{t_k} V_{h+1}^{t_k})(x,a)
$$

$$
\leq (\mathbb{P}_h V_{h+1}^{t_k})(x,a) - \min\{\phi(x,a)^\top w_h^{t_k} + \Gamma_h^{t_k}(x,a), H - h\}
$$

$$
= \max\{(\mathbb{P}_h V_{h+1}^{t_k})(x,a) - \phi(x,a)^\top w_h^{t_k} - \Gamma_h^{t_k}(x,a), (\mathbb{P}_h V_{h+1}^{t_k})(x,a) - (H-h)\},
$$

$$
\leq \max\{0, 0\} = 0, \tag{B.29}
$$

where the last inequality uses (B.28) and the fact that $\|V_{h+1}^{t_k}\|_\infty \leq H - h$. Moreover, by (B.28), we have

$$
(\mathbb{P}_h V_{h+1}^{t_k} - \widehat{\mathbb{P}}_h^{t_k} V_{h+1}^{t_k})(x,a) \geq (\mathbb{P}_h V_{h+1}^{t_k})(x,a) - \phi(x,a)^\top w_h^{t_k} - \Gamma_h^{t_k}(x,a)
$$

$$
\geq -2\Gamma_h^{t_k}(x,a). \tag{B.30}
$$

Combining (B.29), (B.30), and the fact that $(\mathbb{P}_h V_{h+1}^{t_k} - \widehat{\mathbb{P}}_h^{t_k} V_{h+1}^{t_k})(\cdot,\cdot) \geq -2H$, we obtain

$$
-2\min\{H, \Gamma_h^{t_k}(x,a)\} \leq (\mathbb{P}_h V_{h+1}^{t_k} - \widehat{\mathbb{P}}_h^{t_k} V_{h+1}^{t_k})(x,a) \leq 0,
$$

which concludes the proof of Lemma 4.4. $\qquad\square$

## B.5 Proof of Lemma 4.5

*Proof.* For ease of presentation, we define

$$
\Gamma_h^k(\cdot,\cdot) = \beta \sqrt{\phi(\cdot,\cdot)^\top (\Lambda_h^k)^{-1} \phi(\cdot,\cdot)}, \qquad \Lambda_h^k = \sum_{\tau=1}^{k-1} \phi(x_h^\tau, a_h^\tau) \phi(x_h^\tau, a_h^\tau)^\top + \lambda I_d, \tag{B.31}
$$

for all $(k,h) \in [K] \times [H]$. Then we have the following lemma, which uses the elliptical potential to bound $\sum_{k=1}^K \sum_{h=1}^H \Gamma_h^k(x_h^k, a_h^k)$.

**Lemma B.4.** It holds that

$$\sum_{k=1}^{K} \sum_{h=1}^{H} \Gamma_h^k(x_h^k, a_h^k) \leq \mathcal{O}(d^{3/4} H^2 K^{3/4} \cdot \iota).$$

*Proof.* By the definition of $\Gamma_h^k$ in (B.31), we have

$$\sum_{k=1}^{K} \sum_{h=1}^{H} \Gamma_h^k(x_h^k, a_h^k) = \beta \sum_{k=1}^{K} \sum_{h=1}^{H} \sqrt{\phi(x_h^k, a_h^k)^\top (\Lambda_h^k)^{-1} \phi(x_h^k, a_h^k)}. \tag{B.32}$$

Applying Cauchy-Schwarz inequality to (B.32), we have

$$\sum_{k=1}^{K} \sum_{h=1}^{H} \Gamma_h^k(x_h^k, a_h^k) \leq \beta \sum_{h=1}^{H} \left( K \cdot \sum_{k=1}^{K} \phi(x_h^k, a_h^k)^\top (\Lambda_h^k)^{-1} \phi(x_h^k, a_h^k) \right)^{1/2}$$

$$\leq \beta \sum_{h=1}^{H} \left[ 2K \cdot \log \left( \frac{\det(\Lambda_h^{K+1})}{\det(\Lambda_h^1)} \right) \right]^{1/2}, \tag{B.33}$$

where the last inequality uses the elliptical potential lemma (Lemma D.5). For any $h \in [H]$, we have

$$\Lambda_h^1 = \lambda \cdot I_d, \qquad \Lambda_h^{K+1} = \sum_{k=1}^{K} \phi(x_h^k, a_h^k)\phi(x_h^k, a_h^k)^\top + \lambda I_d \preceq (K + \lambda) \cdot I_d, \tag{B.34}$$

where the inequality uses the fact that $\|\phi(\cdot, \cdot)\|_2 \leq 1$. Plugging (B.34) into (B.33), we have

$$\sum_{k=1}^{K} \sum_{h=1}^{H} \Gamma_h^k(x_h^k, a_h^k) \leq H\beta \cdot \left[ 2dK \cdot \log \left( \frac{K + \lambda}{\lambda} \right) \right]^{1/2}.$$

Together with the facts that $\lambda = 1$, $\iota = \log(dHK|\mathcal{A}|/\delta)$ with $\delta \in (0, 1]$, and $\beta = \mathcal{O}(d^{1/4} H K^{1/4} \iota^{1/2})$, we conclude the proof of Lemma B.4. $\square$

We also need the following lemma to connect the quantity $\sum_{k=1}^{K} \sum_{h=1}^{H} \Gamma_h^k(x_h^k, a_h^k)$ in Lemma B.4 and our target $\sum_{k=1}^{K} \sum_{h=1}^{H} \min\{H, \Gamma_h^{t_k}(x_h^k, a_h^k)\}$.

**Lemma B.5.** For any $h \in [H]$, we define the set $\mathcal{E}_h$ as

$$\mathcal{E}_h = \{k : \Gamma_h^{t_k}(x_h^k, a_h^k)/\Gamma_h^k(x_h^k, a_h^k) > 2\}. \tag{B.35}$$

Then we have

$$|\mathcal{E}_h| \leq \mathcal{O}(d^{5/2} K^{1/2} \cdot \iota).$$

*Proof.* For $k \in \mathcal{E}_h$, there exists $i \in [L]$ such that $k_i \leq k < k_{i+1}$. Then we know $t_k = k_i$, which further implies that

$$\log \left( \frac{\det(\Lambda_h^{k_{i+1}})}{\det(\Lambda_h^{k_i})} \right) \geq \log \left( \frac{\det(\Lambda_h^k)}{\det(\Lambda_h^{k_i})} \right) \geq \log \left( \frac{\phi(x_h^k, a_h^k)^\top (\Lambda_h^{k_i})^{-1} \phi(x_h^k, a_h^k)}{\phi(x_h^k, a_h^k)^\top (\Lambda_h^k)^{-1} \phi(x_h^k, a_h^k)} \right)$$

$$= 2 \log \left( \frac{\Gamma_h^{k_i}(x_h^k, a_h^k)}{\Gamma_h^k(x_h^k, a_h^k)} \right),$$

where the first inequality follows from the fact that $\Lambda_h^{k_{i+1}} \succeq \Lambda_h^k$, the second inequality uses Lemma D.6, and the last equality is obtained by the definitions of $\Gamma_h^k$ and $\Gamma_h^{k_i}$ in (B.31). Together with the definition of $\mathcal{E}_h$ in (B.35), we further obtain that

$$\log \left( \frac{\det(\Lambda_h^{k_{i+1}})}{\det(\Lambda_h^{k_i})} \right) \geq 2 \log 2. \tag{B.36}$$

Meanwhile, we have

$$\sum_{i=1}^{L} \log\left(\frac{\det(\Lambda_h^{k_{i+1}})}{\det(\Lambda_h^{k_i})}\right) = \log\left(\frac{\det(\Lambda_h^{K+1})}{\det(\Lambda_h^1)}\right) \le d \cdot \log\left(\frac{K+\lambda}{\lambda}\right), \tag{B.37}$$

where the last inequality follows from the facts that

$$\Lambda_h^1 = \lambda \cdot I_d, \qquad \Lambda_h^{K+1} = \sum_{k=1}^{K} \phi(x_h^k, a_h^k)\phi(x_h^k, a_h^k)^\top + \lambda I_d \preceq (K+\lambda) \cdot I_d.$$

Here the last inequality uses $\|\phi(\cdot,\cdot)\|_2 \le 1$. Combining (B.36) and (B.37), we have

$$\left|\left\{i \in [L] : \log\left(\frac{\det(\Lambda_h^{k_{i+1}})}{\det(\Lambda_h^{k_i})}\right)\right\}\right| \le \frac{d\log\left((K+\lambda)/\lambda\right)}{2\log 2}. \tag{B.38}$$

Since each batch contains $B$ episodes, we obtain that

$$|\mathcal{E}_h| \le B \cdot \left|\left\{i \in [L] : \log\left(\frac{\det(\Lambda_h^{k_{i+1}})}{\det(\Lambda_h^{k_i})}\right)\right\}\right| \le \mathcal{O}(d^{5/2}K^{1/2} \cdot \iota),$$

where the last inequality uses $\lambda = 1$, $\iota = \log(dHK|\mathcal{A}|/\delta)$, $B = \sqrt{d^3 K}$ and (B.38). Therefore, we conclude the proof of Lemma B.5. $\qquad\square$

Back to our proof of Lemma 4.5, we have

$$\sum_{k=1}^{K}\sum_{h=1}^{H} \min\{H, \Gamma_h^{t_k}(x_h^k, a_h^k)\} = \sum_{h=1}^{H}\sum_{k\in\mathcal{E}_h} \min\{H, \Gamma_h^{t_k}(x_h^k, a_h^k)\} + \sum_{h=1}^{H}\sum_{k\notin\mathcal{E}_h} \min\{H, \Gamma_h^{t_k}(x_h^k, a_h^k)\}. \tag{B.39}$$

For any $h \in [H]$ and $k \notin \mathcal{E}_h$, by the definition of $\mathcal{E}_h$, we have

$$\min\{H, \Gamma_h^{t_k}(x_h^k, a_h^k)\} \le \min\{H, 2\Gamma_h^k(x_h^k, a_h^k)\} \le 2\Gamma_h^k(x_h^k, a_h^k). \tag{B.40}$$

Combining (B.39), (B.40), and the fact that $\min\{H, \Gamma_h^{t_k}(x_h^k, a_h^k)\} \le H$, we have

$$\sum_{k=1}^{K}\sum_{h=1}^{H} \min\{H, \Gamma_h^{t_k}(x_h^k, a_h^k)\} \le H\sum_{h=1}^{H} |\mathcal{E}_h| + \sum_{k=1}^{K}\sum_{h=1}^{H} \Gamma_h^k(x_h^k, a_h^k)$$
$$\le \mathcal{O}(d^{3/4}H^2K^{3/4} \cdot \iota) + \mathcal{O}(d^{5/2}H^2K^{1/2} \cdot \iota),$$

where the last inequality uses Lemmas B.4 and B.5. Therefore, we conclude the proof of Lemma 4.5.
$\qquad\square$

## C   Proof for Concentration of Self-Normalized Processes

*Proof of Lemma B.3.* By the previous concentration lemma of self-normalized process (Lemma D.3), for any $\delta' \in (0, 1]$, $\varepsilon > 0$, and $(k, h) \in [K] \times [H]$ we have

$$\left\|\sum_{\tau=1}^{t_k-1} \phi(x_h^\tau, a_h^\tau) \cdot \left(V_{h+1}^{t_k}(x_{h+1}^\tau) - (\mathbb{P}_h V_{h+1}^{t_k})(x_h^\tau, a_h^\tau)\right)\right\|_{(\Lambda_h^{t_k})^{-1}}^2$$
$$\le 4H^2 \cdot \left[\frac{d}{2}\log\left(\frac{k+\lambda}{\lambda}\right) + \log\left(\frac{\mathcal{N}_\varepsilon(\mathcal{V}_{h+1}^k)}{\delta'}\right)\right] + \frac{8k^2\varepsilon^2}{\lambda} \tag{C.1}$$

with probability $1 - \delta'$. Here $\mathcal{N}_\varepsilon(\mathcal{V}_{h+1}^k)$ is the $\varepsilon$-covering number of function class $\mathcal{V}_{h+1}^k$, which is defined by

$$\mathcal{V}_{h+1}^k = \left\{V(\cdot) = \langle Q(\cdot,\cdot), \pi(\cdot\,|\,\cdot)\rangle : Q \in \mathcal{Q}_{h+1}^k, \pi \in \Pi_{h+1}^k\right\}, \tag{C.2}$$

where $\mathcal{Q}_{h+1}^k$ and $\Pi_{h+1}^k$ are Q-function class and policy class, respectively. Specifically, $\mathcal{Q}_{h+1}^k$ is a function class with the following parametric form

$$\mathcal{Q}_{h+1}^k = \big\{ \bar{r}_h^{t_k} + \{\phi^\top w + \beta \cdot (\phi^\top \Lambda^{-1} \phi)^{1/2}, H - h\}^+ : \|w\|_2 \leq H\sqrt{dK/\lambda}, \lambda_{\min}(\Lambda) \geq \lambda \big\}. \tag{C.3}$$

Here we uses $\|w_{h+1}^k\|_2 \leq H\sqrt{dK/\lambda}$ (Lemma D.1) and $\lambda_{\min}(\Lambda_{h+1}^k) \geq \lambda$ to ensure that $Q_{h+1}^k \in \mathcal{Q}_{h+1}^k$. Meanwhile, the policy class $\Pi_{h+1}^k$ is defined as

$$\Pi_{h+1}^k = \Big\{ \pi(\cdot \mid \cdot) \propto \exp\Big(\sum_{i=1}^l \alpha Q_i(\cdot, \cdot)\Big) : Q_i \in \mathcal{Q}_{h+1}^{k_i}, \forall i \in [l] \Big\}, \quad \text{where } l = \max\{i : k_i < t_k\}. \tag{C.4}$$

Notable, here $l \leq L$ since our algorithm only has $L$ batches. Also, $\pi_{h+1}^{t_k}$ in (C.2) belongs to this policy class since it takes the following form

$$\pi_{h+1}^{t_k}(a \mid x) = \frac{\exp(\sum_{i=1}^l \alpha Q_{h+1}^{k_i}(x, a))}{\sum_{a' \in \mathcal{A}} \exp(\sum_{i=1}^l \alpha Q_{h+1}^{k_i}(x, a'))},$$

For policy class defined in (C.4), we define its covering number with respect to the following distance

$$\text{dist}(\pi, \pi') = \sup_{x \in \mathcal{S}} \|\pi(\cdot \mid x) - \pi'(\cdot \mid x)\|_1.$$

The following lemma connects the covering number of value class in (C.2) to covering numbers of the Q-function class in (C.3) and the policy class in (C.4).

**Lemma C.1.** It holds that

$$\mathcal{N}_\varepsilon(\mathcal{V}_{h+1}^k) \leq \mathcal{N}_{\varepsilon/2}(\mathcal{Q}_{h+1}^k) \cdot \mathcal{N}_{\varepsilon/(2H)}(\Pi_{h+1}^k).$$

*Proof.* See §C.1 for a detailed proof. $\qquad\square$

Following the standard covering argument (Lemma D.4), we can derive an upper bound for $\mathcal{N}_{\varepsilon/2}(\mathcal{Q}_{h+1}^k)$. However, $\mathcal{N}_{\varepsilon/(2H)}(\Pi_{h+1}^k)$ is relatively difficult to bound since the log-covering number of a $|\mathcal{A}|$-dimensional probability distribution is $\widetilde{\mathcal{O}}(|\mathcal{A}|)$. Fortunately, we have the following lemma that utilizes the structure of policy class (C.4) and converts the covering number of the policy class to the covering numbers of several Q-function classes.

**Lemma C.2.** It holds that

$$\mathcal{N}_{\varepsilon/(2H)}(\Pi_{h+1}^k) \leq \prod_{i=1}^l \mathcal{N}_{\varepsilon^2/(16\alpha l H^2)}(\mathcal{Q}_{h+1}^{k_i})$$

*Proof.* See §C.2 for a detailed proof. $\qquad\square$

Note that $\lambda = 1$, $l \leq L$, $L = K/B$, $B = \sqrt{d^3 K}$, $\alpha = \sqrt{2B \log |\mathcal{A}|/(KH^2)}$, $\iota = \log(dHK|\mathcal{A}|/\delta)$, and $\beta = \mathcal{O}(d^{1/4} H K^{1/4} \iota^{1/2})$. Meanwhile, let $\varepsilon = 1/K$ and $\delta' = \delta/2$. Combining (C.1) with Lemmas C.1, C.2, and D.4, we have

$$\left\| \sum_{\tau=1}^{t_k-1} \phi(x_h^\tau, a_h^\tau) \cdot \big(V_{h+1}^{t_k}(x_{h+1}^\tau) - (\mathbb{P}_h V_{h+1}^{t_k})(x_h^\tau, a_h^\tau)\big) \right\|_{(\Lambda_h^{t_k})^{-1}} \leq \mathcal{O}(\sqrt{d^2 H^2 L \cdot \iota})$$
$$= \mathcal{O}(d^{1/4} H K^{1/4} \iota^{1/2}),$$

which concludes the proof of Lemma B.3. $\qquad\square$

## C.1 Proof of Lemma C.1

*Proof.* Suppose (i) $\mathcal{Q}^k_{\varepsilon/2,h+1}$ is an $\varepsilon/2$-net of $\mathcal{Q}^k_{h+1}$ with $|\mathcal{Q}^k_{\varepsilon/2,h+1}| = \mathcal{N}_{\varepsilon/2}(\mathcal{Q}^k_{h+1})$; and (ii) $\Pi^k_{\varepsilon/(2H),h+1}$ is an $\varepsilon/(2H)$-net of $\Pi^k_{h+1}$ with $|\Pi^k_{\varepsilon/(2H),h+1}| = \mathcal{N}_{\varepsilon/(2H)}(\Pi^k_{h+1})$. Then we show $\mathcal{Q}^k_{\varepsilon/2,h+1} \times \Pi^k_{\varepsilon/(2H),h+1}$ induces an $\varepsilon$-net of $\mathcal{V}^k_{h+1}$, and thus obtaining the desired result. Specifically, for any $V = \langle Q(\cdot,\cdot), \pi(\cdot \mid \cdot) \rangle$ with $(Q,\pi) \in \mathcal{Q}^k_{h+1} \times \Pi^k_{h+1}$, we can find $(Q', \pi') \in \mathcal{Q}^k_{\varepsilon/2,h+1} \times \Pi^k_{\varepsilon/(2H),h+1}$ such that

$$\sup_{(x,a)\in\mathcal{S}\times\mathcal{A}} |Q(x,a) - Q'(x,a)| \leq \varepsilon/2, \qquad \sup_{x\in\mathcal{S}} \|\pi(\cdot \mid x) - \pi'(\cdot \mid x)\|_1 \leq \varepsilon/(2H). \qquad \text{(C.5)}$$

Let $V' = \langle Q'(\cdot,\cdot), \pi'(\cdot \mid \cdot) \rangle$, we have

$$\begin{aligned}
\sup_{x\in\mathcal{S}} |V(x) - V'(x)| &= \sup_{x\in\mathcal{S}} \left| \langle Q(x,\cdot), \pi(\cdot \mid x) \rangle - \langle Q'(x,\cdot), \pi'(\cdot \mid x) \rangle \right| \\
&\leq \sup_{x\in\mathcal{S}} \left| \langle Q(x,\cdot) - Q'(x,\cdot), \pi(\cdot \mid x) \rangle \right| + \sup_{x\in\mathcal{S}} \left| \langle Q'(x,\cdot), \pi(\cdot \mid x) - \pi'(\cdot \mid x) \rangle \right| \\
&\leq \sup_{(x,a)\in\mathcal{S}\times\mathcal{A}} |Q(x,a) - Q'(x,a)| + H \cdot \sup_{x\in\mathcal{S}} \|\pi(\cdot \mid x) - \pi'(\cdot \mid x)\|_1 \\
&\leq \varepsilon/2 + \varepsilon/2 = \varepsilon,
\end{aligned}$$

where the first inequality follows from the triangle inequality, the second inequality uses Cauchy-Schwarz inequality and the fact that $\|Q'\|_\infty \leq H$, and the last inequality follows from (C.5). Therefore, we conclude the proof of Lemma C.1. $\qquad\square$

## C.2 Proof of Lemma C.2

*Proof.* Suppose $\mathcal{Q}^{k_i}_{\varepsilon^2/(16\alpha lH^2),h+1}$ is the minimum $\varepsilon^2/(16\alpha lH^2)$-net of $\mathcal{Q}^{k_i}_{h+1}$ for all $i \in [l]$. Then for any $\pi \propto \exp(\alpha \sum_{i=1}^l Q_i)$ with $Q_i \in \mathcal{Q}^{k_i}_{h+1}$ for all $i \in [l]$, we can choose $\{Q'_i \in \mathcal{Q}^{k_i}_{\varepsilon^2/(16\alpha lH^2),h+1}\}$ such that

$$\sup_{(x,a)\in\mathcal{S}\times\mathcal{A}} |Q_i(x,a) - Q'_i(x,a)| \leq \frac{\varepsilon^2}{16\alpha lH^2}, \qquad \forall i \in [l]. \qquad \text{(C.6)}$$

Hence, we have

$$\sup_{(x,a)\in\mathcal{S}\times\mathcal{A}} \left| \alpha \sum_{i=1}^l Q_i(x,a) - \alpha \sum_{i=1}^l Q'_i(x,a) \right| \leq \alpha \sum_{i=1}^l \sup_{(x,a)\in\mathcal{S}\times\mathcal{A}} |Q_i(x,a) - Q'_i(x,a)| \leq \frac{\varepsilon^2}{16H^2}, \qquad \text{(C.7)}$$

where the first inequality uses the triangle inequality and the last inequality follows from (C.6). Then we use the following lemma to establish the upper bound for $\sup_{x\in\mathcal{S}} \|\pi(\cdot \mid x) - \pi'(\cdot \mid x)\|_1$.

**Lemma C.3.** For $\pi, \pi' \in \Delta(\mathcal{X})$ and $Q, Q' : \mathcal{X} \mapsto \mathbb{R}^+$, if $\pi(\cdot) \propto \exp(Q(\cdot))$ and $\pi'(\cdot) \propto \exp(Q'(\cdot))$, we have

$$\|\pi - \pi'\|_1 \leq 2\sqrt{\|Q - Q'\|_\infty}.$$

*Proof.* See §C.3 for a detailed proof. $\qquad\square$

By Lemma C.3, we have

$$\sup_{x\in\mathcal{S}} \|\pi(\cdot \mid x) - \pi'(\cdot \mid x)\|_1 \leq 2\sqrt{\sup_{(x,a)\in\mathcal{S}\times\mathcal{A}} \left| \alpha \sum_{i=1}^l Q_i(x,a) - \alpha \sum_{i=1}^l Q'_i(x,a) \right|} \leq \frac{\varepsilon}{2H},$$

where the last inequality follows from (C.7). Therefore, we have

$$\mathcal{N}_{\varepsilon/(2H)}(\Pi^k_{h+1}) \leq \prod_{i=1}^l \mathcal{N}_{\varepsilon^2/(16\alpha lH^2)}(\mathcal{Q}^{k_i}_{h+1}),$$

which concludes the proof of Lemma C.2. $\qquad\square$

## C.3 Proof of Lemma C.3

*Proof.* Since $\pi(\cdot) \propto \exp(Q(\cdot))$ and $\pi'(\cdot) \propto \exp(Q'(\cdot))$, we have for any $x \in \mathcal{X}$:

$$\frac{\pi(x)}{\pi'(x)} = \frac{\exp(Q(x))}{\exp(Q'(x))} \cdot \frac{\sum_{x' \in \mathcal{X}} \exp(Q'(x'))}{\sum_{x' \in \mathcal{X}} \exp(Q(x'))}.$$

Note that for any $x \in \mathcal{X}$ we have

$$\begin{cases} \frac{\exp(Q(x))}{\exp(Q'(x))} = \exp(Q(x) - Q'(x)) \leq \exp(\|Q - Q'\|_\infty) \\ \frac{\exp(Q'(x))}{\exp(Q(x))} = \exp(Q'(x) - Q(x)) \leq \exp(\|Q - Q'\|_\infty) \end{cases},$$

which implies that

$$\frac{\pi(x)}{\pi'(x)} \leq \exp(\|Q - Q'\|_\infty) \cdot \frac{\exp(\|Q - Q'\|_\infty) \cdot \sum_{x' \in \mathcal{X}} \exp(Q(x'))}{\sum_{x' \in \mathcal{X}} \exp(Q(x'))} = \exp(2\|Q - Q'\|_\infty).$$

Hence, we have

$$\mathrm{KL}(\pi\|\pi') = \sum_{x \in \mathcal{X}} \pi(x) \log \frac{\pi(x)}{\pi'(x)} \leq 2\|Q - Q'\|_\infty \cdot \sum_{x \in \mathcal{X}} \pi(x) = 2\|Q - Q'\|_\infty.$$

Finally, by Pinsker's inequality, we have

$$\|\pi - \pi'\|_1 \leq \sqrt{2 \cdot \mathrm{KL}(\pi\|\pi')} \leq 2\sqrt{\|Q - Q'\|_\infty},$$

which concludes the proof of Lemma C.3. $\qquad\square$

## D  Auxiliary Lemmas

**Lemma D.1.** For any $(i, h) \in [L] \times [H]$, the linear coefficient $w_h^{k_i}$ defined in Line 12 of Algrotihm 1 satisfies

$$\|w_h^{k_i}\| \leq H\sqrt{dK/\lambda}.$$

*Proof.* Fix $(i, h) \in [L] \times [H]$. Our proof follows the proof of Lemma B.2 in Jin et al. [17]. For any $v \in \mathbb{R}^d$ with $\|v\|_2 = 1$, by the definition of $w_h^{k_i}$ we have

$$|v^\top w_h^{k_i}| = \left| v^\top (\Lambda_h^{k_i})^{-1} \sum_{\tau=1}^{k_i-1} \phi(x_h^\tau, a_h^\tau) \cdot V_{h+1}^{k_i}(x_{h+1}^\tau) \right|.$$

Since $\|V_{h+1}^{k_i}\|_\infty \leq H$, we further have

$$|v^\top w_h^{k_i}| \leq H \sum_{\tau=1}^{k_i-1} \left| v^\top (\Lambda_h^{k_i})^{-1} \phi(x_h^\tau, a_h^\tau) \right|$$

$$\leq H \cdot \left[ \sum_{\tau=1}^{k_i-1} v^\top (\Lambda_h^{k_i})^{-1} v \right]^{1/2} \cdot \left[ \sum_{\tau=1}^{k_i-1} \phi(x_h^\tau, a_h^\tau)^\top (\Lambda_h^{k_i})^{-1} \phi(x_h^\tau, a_h^\tau) \right]^{1/2}$$

$$\leq H\sqrt{dK/\lambda},$$

where the second inequality is obtained by Cauchy-Schwarz inequality, and the last inequality uses $\lambda I_d \preceq \Lambda_h^{k_i}$, $k_i \leq K$, $\|v\|_2 = 1$ and Lemma D.2. Hence, we have

$$\|w_h^{k_i}\|_2 = \sup_{\|v\|_2=1} |v^\top w_h^{k_i}| \leq H\sqrt{dK/\lambda},$$

which concludes the proof of Lemma D.1. $\qquad\square$

**Lemma D.2.** Let $\Lambda_t = \lambda \cdot I_d + \sum_{i=1}^t \phi_i \phi_i^\top$ with $\phi_i \in \mathbb{R}^d$ and $\lambda > 0$. Then we have

$$\sum_{i=1}^t \phi_i (\Lambda_t)^{-1} \phi_i \leq d.$$

*Proof.* See Lemma D.1 in Jin et al. [17] for a detailed proof. □

**Lemma D.3.** Let $\{x_\tau \in \mathcal{S}\}_{\tau=1}^\infty$ and $\{\phi_\tau \in \mathbb{R}^d\}_{\tau=1}^\infty$ with $\|\phi_\tau\|_2 \leq 1$ be stochastic processes adapted to the filtration $\{\mathcal{F}_\tau\}_{\tau=1}^\infty$. Let $\Lambda_k = \sum_{\tau=1}^{k-1} \phi_\tau \phi_\tau^\top + \lambda \cdot I_d$. Then for any $\delta \in (0,1]$, with probability at least $1 - \delta$, for all $k \in \mathbb{N}^+$, and function $V \in \mathcal{V}$ satisfying $\sup_{x\in\mathcal{S}} |V(s)| \leq H$, we have

$$\left\| \sum_{\tau=1}^{k-1} \phi_\tau \big(V(x_{\tau+1}) - \mathbb{E}[V(x_{\tau+1}) \mid \mathcal{F}_\tau]\big) \right\|_{\Lambda_k^{-1}}^2 \leq 4H^2 \cdot \left[ \frac{d}{2} \log\left(\frac{k+\lambda}{\lambda}\right) + \log\left(\frac{\mathcal{N}_\varepsilon}{\delta}\right) \right] + \frac{8k^2\varepsilon^2}{\lambda},$$

where $\mathcal{N}_\varepsilon$ is the $\varepsilon$-covering number of the function class $\mathcal{V}$ with respect to the distance $\mathrm{dist}(V, V') = \sup_{x\in\mathcal{S}} |V(x) - V'(x)|$.

*Proof.* See Lemma D.4 of Jin et al. [17] for a detailed proof. □

**Lemma D.4.** For any $h \in [H]$, let $\mathcal{Q}_h$ be a function class mapping from $\mathcal{S} \times \mathcal{A}$ to $\mathbb{R}$ with the form

$$\mathcal{Q}_h(\cdot, \cdot) = r(\cdot, \cdot) + \min\left\{ \phi(\cdot, \cdot)^\top w + \beta\sqrt{\phi(\cdot, \cdot)^\top \Lambda^{-1} \phi(\cdot, \cdot)}, H - h \right\}^+,$$

where $r : \mathcal{S} \times \mathcal{A} \mapsto [0,1]$, $\|w\|_2 \leq L$, $\lambda_{\min}(\lambda) \geq \lambda$. Assuming $\|\phi(\cdot, \cdot)\|_2 \leq 1$ and $\beta > 0$, we have

$$\log \mathcal{N}_\varepsilon(\mathcal{Q}_h) \leq d \log(1 + 4L/\varepsilon) + d^2 \log\left(1 + 8d^{1/2}\beta^2/(\lambda\varepsilon)\right),$$

where $\mathcal{N}_\varepsilon(\mathcal{Q}_h)$ is the $\varepsilon$-covering number of the function class $\mathcal{Q}_h$ with respect to the distance $\mathrm{dist}(Q, Q') = \sup_{(x,a)\in\mathcal{S}\times\mathcal{A}} |Q(x, a) - Q'(x, a)|$.

*Proof.* See Lemma D.6 of Jin et al. [17] for a detailed proof. □

**Lemma D.5** (Elliptical Potential Lemma). Let $\{\phi_t \in \mathbb{R}^d\}_{t=1}^\infty$ satisfy $\|\phi_t\|_2 \leq 1$ for all $t \in \mathbb{N}^+$. Moreover, let $\Lambda_0 \in \mathbb{R}^{d\times d}$ be a positive-definite matrix with $\lambda_{\min}(\Lambda_0)$ and $\Lambda_t = \Lambda_0 + \sum_{i=1}^{t-1} \phi_i \phi_i^\top$ for any $t \in \mathbb{N}^+$. Then for any $t \in \mathbb{N}^+$, we have

$$\log\left(\frac{\det(\Lambda_{t+1})}{\det(\Lambda_1)}\right) \leq \sum_{i=1}^t \phi_i^\top \Lambda_i^{-1} \phi_i \leq 2\log\left(\frac{\det(\Lambda_{t+1})}{\det(\Lambda_1)}\right).$$

*Proof.* See Lemma 11 of Abbasi-Yadkori et al. [1] for a detailed proof. □

**Lemma D.6.** Suppose $\Lambda, \Lambda' \in \mathbb{R}^{d\times d}$ are two positive definite matrices and satisfy $\Lambda \preceq \Lambda'$, then for any $x \in \mathbb{R}^d$, we have

$$\frac{\det(\Lambda')}{\det(\Lambda)} \geq \frac{x^\top \Lambda' x}{x^\top \Lambda x}.$$

*Proof.* See Lemma 12 of Abbasi-Yadkori et al. [1] for a detailed proof. □

