# OpenReview forum: "A Theoretical Analysis of Optimistic Proximal Policy Optimization in Linear Markov Decision Processes"
_NeurIPS.cc/2023/Conference — NeurIPS 2023 poster_

### Official Review · Reviewer_Eaqc · 2023-06-26

**Soundness:** 3 good
**Presentation:** 4 excellent
**Contribution:** 3 good
**Rating:** 6
**Confidence:** 3

**Summary:**

In this paper, the authors extend the theory of proximal policy optimization-based methods in the linear mixture MDPs and propose an optimistic variant PPO algorithm (OPPO+) for stochastic linear MDPs and adversarial linear MDPs with full information.
The proposed algorithm adopts a multi-batched updating rule from bandit literature, where the policy is updated at regular intervals of batches, instead of updating it every episode. During the policy improvement step, the policy is updated using proximal policy optimization, while the policy evaluation step involves estimating the value using least square value iteration with the average reward from the previous batch.
The proposed algorithm provides a regret guarantee for $\tilde{O}(d^{3/4} H^2 K^{3/4})$ and has a sample complexity of $\tilde{O}(d^3H^8/\epsilon^4 + d^5H^4/\epsilon^2)$. This regret bound is tighter than the existing PPO-based algorithms proposed for stochastic linear MDPs.

**Strengths:**

- The proposed algorithm can be applied not only to stochastic linear MDPs but also to adversarial linear MDPs with full-information feedback.
In order to cover the case of adversarial linear MDPs, the average reward technique is used, and for the analysis, result for bound on the gap between the value of the policy from the previous batch and the value of the policy from the current batch is suggested.
- When applying the existing covering number theory from linear MDPs, the regret may depend on the size of the action space. To address this, a method using novel covering theory is proposed to reduce it to a logarithmic scale.
- Compared to existing policy optimization algorithms, it provides a tighter regret guarantee in both the stochastic linear MDP and adversarial linear MDP with full-information feedback settings.

**Weaknesses:**

- The computational cost of policy improvement is not discussed in detail.
- It remains unclear what advantages the proposed policy optimization algorithms offer compared to value-based algorithms, apart from their theoretical purposes.

**Questions:**

1. What does it mean for the proposed algorithm to be "optimistic"? What I mean is that for example, in value-based algorithms, optimism refers to the estimated values of the algorithm being more optimistic than the true optimal value (Lemma B.5 in [17]).
2. Can you explain the computational cost of policy improvement in OPPO+ compared to the policy improvement in value-based algorithms [17]?
3. Besides the ability to learn stochastic policies, what other advantages does OPPO+ have compared to value-based algorithms?
4. In line 203, the author argued that value-based algorithms cannot handle adversarial rewards. Could you provide more details about this?
5. By performing policy optimization infrequently, the algorithm achieves a regret of order $O(K^{3/4})$. What are the drawbacks or trade-offs associated with this approach? In other words, in line 81, it was mentioned that if policy optimization is performed every episode, the regret would have a linear dependence on $K$. What benefits can be obtained from this approach?

**Limitations:**

As this is a theoretical paper, it does not seem to have any negative societal impact. However, the authors have not mentioned the limitations of this algorithm. (If they have mentioned any, please let me know, and I will check.) Nonetheless, they have provided an introduction to future work in Remark 3.3.

---

> ### Author Rebuttal · Authors · 2023-08-08
>
> Thanks for your review and positive feedback. We will try to address your concerns in the following.
>
> **Q1:** What does it mean for the proposed algorithm to be "optimistic"? What I mean is that for example, in value-based algorithms, optimism refers to the estimated values of the algorithm being more optimistic than the true optimal value (Lemma B.5 in [17]).
>
> **A1:** Thanks for your good question, this is indeed one of the most important differences between policy-based algorithms and value-based algorithms.
> - For the value-based algorithm LSVI-UCB in [17], they can obtain that $V_1^* \le V_1^k$. Furthermore,
> $$\mathrm{Regret}(K) = \sum_{k=1}^K V_1^* - V_1^{\pi^k} \le \sum_{k=1}^K V_1^k - V_{1}^{\pi^k} \lesssim \text{sum of bonus function} \le \tilde{\mathcal{O}}(\sqrt{K}).$$
> - For policy-based algorithms, we cannot obtain the guarantee that $V_1^* \le V_1^k$ as Lemma B.5 in [17]. Instead, we can obtain that $V_1^{\pi^k} \le V_1^k$ for stochastic linear MDPs. In other words, $V_1^k$ is the optimistic estimation of the value of policy $\pi^k$. Hence, we cannot follow the analysis way like [17]. Instead, we use the regret decomposition lemma (Lemma 4.1) to decompose the regret into two terms --- policy optimization error and estimation error, and then tackle these two terms separately.
>
> In terms of algorithm design, our algorithm shares the same spirit with LSVI-UCB [17]. In both [17] and our work, the bonus function $\Gamma_h^k$ serves to quantify estimation error (cf. Lemma 4.4 in our paper and Lemma B.4 in [17]). Furthermore, both OPPO+ and LSVI-UCB calculate the "optimistic" estimation by adding bonus functions to the estimated value function (cf. Line 15 of OPPO+ and Line 6 in LSVI-UCB). As a result, OPPO+ and LSVI-UCB can be regarded as the optimistic variant of PPO and LSVI, respectively.
>
>
>
>
> **Q2:** Can you explain the computational cost of policy improvement in OPPO+ compared to the policy improvement in value-based algorithms [17]?
>
> **A2:** OPPO+ updates the policy by solving a proximal policy optimization problem, while LSVI-UCB in [17] simply executes the greedy policy with respect to the estimated value function. Hence, it is hard to say that the computational cost of a single policy optimization in OPPO is lower than one step policy improvement in  LSVI-UCB. However, it is worth noting that OPPO+ adopts the multi-batched updating rule, which leads to better computational efficiency compared with LSVI-UCB that updates policy in each episode.
>
>
> **Q3:** Besides the ability to learn stochastic policies, what other advantages does OPPO+ have compared to value-based algorithms?
>
> **A3:**  OPPO+ can tackle adversarial rewards, while value-based algorithms (e.g., LSVI-UCB) cannot. Further elaboration on this point is provided in detail in **A4**. Moreover, PPO is one of the most widely recognized RL algorithms and linear MDP is arguably the most fundamental RL models with function approximation. Consequently, understanding the theoretical performance of PPO in linear MDPs is important.
>
> **Q4:** In line 203, the author argued that value-based algorithms cannot handle adversarial rewards. Could you provide more details about this?
>
> **A4:** In value-based algorithms, the learner estimates the Q-function and then chooses the greedy policy with respect to the estimated Q-function (see e.g., LSVI-UCB). Meanwhile, it is known to us that the deterministic policy policies will incur linear regret even for adversarial linear bandits (simplified problem of adversarial linear MDPs). Please see Chapter 11 (Exercise 11.2) in [1] for more details.
>
> [1] Lattimore, T. and Szepesvari, C. (2020). Bandit algorithms. Cambridge University Press.
>
> **Q5:** By performing policy optimization infrequently, the algorithm achieves a regret of order $O(K^{3/4})$. What are the drawbacks or trade-offs associated with this approach? In other words, in line 81, it was mentioned that if policy optimization is performed every episode, the regret would have a linear dependence on $K$. What benefits can be obtained from this approach?
>
> **A5:** If OPPO+ performs the policy optimization for $L$ times, then by Lemma C.3, we have the complexity (log covering number) of policy class is roughly $\tilde{\mathcal{O}}(L)$ (ignoring the dependency of $d$ and $H$). Furthermore, by the new self-normalized analysis in Appendix C, we have $\beta = \tilde{\mathcal{O}}(\sqrt{L})$, which further implies that the model estimation error is bounded by $\beta \cdot \sum_{k=1}^K\sum_{h=1}^H\sqrt{\phi(x_h^k, a_h^k)(\Lambda_h^k)^{-1}\phi(x_h^k, a_h^k)} \le \tilde{\mathcal{O}}(\sqrt{L K})$ (cf. Lemma 4.5). Meanwhile, by Lemma 4.2, we have the policy optimization error is bounded by $\tilde{\mathcal{O}}(K/\sqrt{L}) = \tilde{\mathcal{O}}(K^{3/4})$. By choosing $L = \Theta(\sqrt{K})$, we obtain a regret of order $\tilde{\mathcal{O}}(K^{3/4})$.
>
> In summary, the multi-batched updating mechanism is crucial to obtain a sublinear regret and the choice of the number of batches is optimal based on our current analysis.

---

> > ### Comment · Reviewer_Eaqc · 2023-08-15
> >
> > Thank you for providing the detailed clarification. It is helpful in understanding the paper. As a result, I have adjusted the scores accordingly.

---

> > > ### Author Response · Authors · 2023-08-15
> > >
> > > Thank you for dedicating your time and providing your valuable support. We will further polish our paper according to your valuable suggestion.

---

### Official Review · Reviewer_r8ZK · 2023-07-03

**Soundness:** 3 good
**Presentation:** 3 good
**Contribution:** 3 good
**Rating:** 7
**Confidence:** 3

**Summary:**

The paper proposes a theoretical analysis of the PPO algorithm. Some novel techniques such as batch-wise update are proposed so the algorithm can work on the adversarial setting of linear MDPs. A regret bound is given, which is better then or comparable to  previous results.

**Strengths:**

The paper is well written, with a clear presentation and is easy to follow. The topic studied in the paper is important. The proposed technique is novel and the theoretical result is solid.

**Weaknesses:**

1. See Question 1. One concern is the similarity between the proposed method and NPG. And additional comparison is needed so the readers could understand the novelty and contribution of the paper more easily.

2. The algorithm design of OPPO+ is similar to OPPO. The difference is OPPO+ replaces the step-wise update with a multi-batched update, and considers the linear MDP setting so the state-action feature $\phi$ can be directly obtained instead of doing integration. The authors claimed the weakness of OPPO, or linear mixture MDP, is that the integration is computationally expensive. But I would rather just regard it as a separate model setting, instead of a weakness. Therefore, when doing comparisons with previous work, besides the literature on linear MDP, it could be great if the authors can also give a more detailed explanation on the setting of linear mixture MDP. In particular, given that the algorithm formulations are so similar, it might be helpful if the author can explain why it's not straightforward to migrate the proof of OPPO to the linear MDP setting.  For example, it can be added to the challenge & novelty section, which will greatly strengthen their argument.

**Questions:**

1. The updating rule of PPO has a very similar formula to Natural Policy Gradient (NPG). There also have been literatures that apply NPG to linear MDPs, can the authors make a comparison between their approach and these literatures?

For example, in section 4.2 of [1], the authors applied NPG to linear MDPs, and obtained a convergence rate of order $d^2H^6/\varepsilon^3$, which should imply a regret bound of order $d^{1/2}H^{3/2}K^{3/4}$.

[1] Liu et al., Optimistic Natural Policy Gradient: a Simple Policy Optimization Algorithm for Online Learning in Linear MDPs, https://arxiv.org/pdf/2305.11032.pdf

2. Is it possible to extend the analysis to the general function approximation setting?

3. One novelty of the paper is its ability to handle adversarial rewards. Can the authors explain which part of their algorithm is crucial to achieving this goal? Still take [1] as an example, I think the algorithms have similar formulation, and the difference of [1] is it doesn't use a multi-batched update. Does that mean the multi-batched updating rule is the crucial part for the adversarial setting?

**Limitations:**

Since the paper is focused on a theoretical side, it's unlikely to have potential negative social impact. And some limitations and future research directions are mentioned in the Conclusion section.

---

> ### Author Rebuttal · Authors · 2023-08-08
>
> Thanks for your review and positive feedback. We will try to address your concerns in the following.
>
> **Q1:** Comparison with OPPO for linear mixture MDPs.
>
> **A1:** We agree that linear mixture MDPs and linear MDPs are two different types of MDPs with linear function approximation. We have discussed some related works on linear mixture MDPs in Appendix A. Here we provide a more detailed comparison between linear MDPs and linear mixture MDPs from the perspective of parameter size. For linear mixture MDPs, the transition kernel takes the form $\mathcal{P}_h(s' \mid s, a) = \psi(s, a, s')^\top \beta_h$ with some $\beta_h \in \mathbb{R}^d$. This means that the model of linear mixture MDPs is characterized by $d$ parameters. In contrast, the number of model parameters of linear MDPs scale with the number of states (the number of model parameters is $d \cdot |\mathcal{S}|$ since $\mu_h(s) \in \mathbb{R}^d$ for all $s\in\mathcal{S}$). Therefore, from our perspective, learning linear MDPs is harder than learning linear mixture MDPs.
>
> Regarding the challenges involved in extending OPPO to linear MDPs, we provided a brief explanation in Challenge 1 of Section 1.1. Now, let's delve into a more detailed explanation of these technical challenges.
>
>
> Technically, for linear mixture MDPs (Equation (B.20) in OPPO paper (Cai et al., 2020)), they need to analyze
> $$\bigg\|  \sum\_{\tau = 1}^{k - 1} \phi\_h^\tau(x\_h^\tau, a\_h^\tau) \cdot \big( V\_{h+1}^{\color{red}{\tau}}(x\_{h+1}^\tau) - (\mathbb{P}\_h V\_{h+1}^{\color{red}{\tau}})(x\_h^\tau, a\_h^\tau) \big) \bigg\|\_{(\Lambda\_h^{k})^{-1}}.$$
> Since $V_{h+1}^\tau$ is adapted to $\mathcal{F}\_{k,h,1} = \{(x\_{i}^{\tau}, a\_{i}^{\tau})\}\_{(\tau, i) \in[k-1] \times[H]} \cup\{r^{\tau}\}\_{\tau \in [k]} \cup\{(x\_{i}^{k}, a\_{i}^{k})\}\_{i \in[h]}$, they can bound this term with classical self-normalized process analysis directly (see Lemma D.1 in (Cai et al., 2020)).
>
> In contrast, for linear MDPs (see e.g., (B.26) in our paper or Lemma B.3 in [1]), we need to bound the term
> $$\bigg\|  \sum\_{\tau = 1}^{k - 1} \phi(x\_h^\tau, a\_h^\tau) \cdot \big( V\_{h+1}^{\color{red}{k}}(x\_{h+1}^\tau) - (\mathbb{P}\_h V\_{h+1}^{\color{red}{k}})(x\_h^\tau, a\_h^\tau) \big) \bigg\|\_{(\Lambda\_h^{k})^{-1}}.$$
> Since $V\_{h+1}^\tau$ is NOT adapted to $\mathcal{F}\_{k,h,1} = \{(x_{i}^{\tau}, a_{i}^{\tau})\}\_{(\tau, i) \in[k-1] \times[H]} \cup \{r^{\tau}\}\_{\tau \in[k]} \cup\{(x_{i}^{k}, a_{i}^{k})\}\_{i \in[h]}$, we need to perform the uniform concentration on the function class of $V\_{h+1}^k$. The challenge of calculating the covering number of this function class has been elaborated in Challenge 1 of Section 1.1 (Lines 58-73).
>
>
> Thanks for your question, and we will add these discussions in the revision.
>
> [1] Provably Efficient Reinforcement Learning with Linear Function Approximation. Chi Jin, Zhuoran Yang, Zhaoran Wang, Michael I. Jordan
>
>
> **Q2:** Comparison with NPG algorithm [2].
>
>
> [2] Liu et al., Optimistic Natural Policy Gradient: a Simple Policy Optimization Algorithm for Online Learning in Linear MDPs, https://arxiv.org/pdf/2305.11032.pdf
>
>
> **A2** Thank you for pointing out this work. Firstly, it is important to note that this work has been released subsequent to the NeurIPS submission deadline. While we have discussed prior policy-based algorithms within our related works, we remain open to discussing the distinctions from [2].
>
> - Regarding the updating rule, both PPO and NPG share a similar policy updating rule. However, we additionally introduce the multi-batched updating and average reward policy evaluation mechanisms, which are crucial to handle adversarial rewards.
> - Regarding the results. Yes, their result implies an $\tilde{\mathcal{O}}(d^{1/2}H^{3/2}K^{3/4})$ regret for *stochastic* linear MDPs. However, their algorithm cannot tackle adversarial linear MDPs, which is the central focus of our paper.
>
>
> **Q3:** Is it possible to extend the analysis to the general function approximation setting?
>
>
> **A3** Yes, our analysis is ready to be extended to the kernel and neural setting [3]. Also, we believe we can deal with low eluder dimension setting like [2] and [4]. Thanks for your question and we will add more discussions in the revision.
>
> [3] Yang, Z., Jin, C., Wang, Z., Wang, M. and Jordan, M. I. (2020). On function approximation in reinforcement learning: Optimism in the face of large state spaces. arXiv preprint
> arXiv:2011.04622.
>
> [4] Reinforcement Learning with General Value Function Approximation: Provably Efficient Approach via Bounded Eluder Dimension
> Ruosong Wang, Ruslan Salakhutdinov, Lin F. Yang
>
> **Q4:** One novelty of the paper is its ability to handle adversarial rewards. Can the authors explain which part of their algorithm is crucial to achieving this goal? Still take [2] as an example, I think the algorithms have similar formulation, and the difference of [2] is it doesn't use a multi-batched update. Does that mean the multi-batched updating rule is the crucial part for the adversarial setting?
>
> **A4:** Compared with [2], we think the average reward policy evaluation and the corresponding analysis are the key to handling adversarial rewards (cf. Challenge 2 and Novelty 2 in Lines 87-102), though the multi-batched updating rule is also important to obtain a sublinear regret. If we use the instantaneous reward to perform policy evaluation like OPPO (Cai et al., 2020) or optimistic NPG in [2], we will suffer the linear regret (cf. Challenge 2 in Lines 87-94). To this end, we use the average reward to evaluate the policies (cf. Lines 95-97). This mechanism will introduce additional errors, which require a new smoothness analysis (cf. Lines 98-102 and Lemma 4.3).

---

> > ### Comment · Reviewer_r8ZK · 2023-08-14
> >
> > Thank you for your response. I have raised the score accordingly.

---

> > > ### Author Response · Authors · 2023-08-14
> > >
> > > Thank you for your review and support. We will further polish our paper according to your valuable suggestion.

---

### Official Review · Reviewer_c9AM · 2023-07-05

**Soundness:** 3 good
**Presentation:** 3 good
**Contribution:** 3 good
**Rating:** 6
**Confidence:** 3

**Summary:**

This paper studies the theoretical performance of an optimistic variant of PPO in episodic adversarial linear MDPs with full-information feedback (i.e., without assuming the reward functions are linear in the feature map), and establishes a regret bound of O(d^3/4 H^2 K^3/4) that matches the optimal regret bound in both stochastic linear MDPs and adversarial linear MDPs. The authors also introduce a new multi-batched updating mechanism to enable a new covering number argument of value and policy classes in their theoretical analysis.

**Strengths:**

Existing theoretical studies of PPO mainly focus on linear mixture MDPs with full-information feedback, which are implemented in a model-based manner and require an integration of the individual base model. In comparison, this work studies another set of linear MDPs that have low-rank representations and proposes a new optimistic variant of PPO that is provably efficient in both stochastic linear MDPs and adversarial linear MDPs.

In particular, this work exhibits several promising results:
1. From the algorithmic perspective, the proposed algorithm involves the novel design of a multi-batched updating mechanism and a policy evaluation step via average rewards.

2. Regarding the performance guarantee, the authors establish the optimal regret bound of O(d^3/4 H^2 K^3/4) with two fundamental findings. Instead of using the existing covering argument in linear MDPs, this work presents a new covering number argument for the value and policy classes. In addition, to ensure the sublinear regret, careful analysis has been done to analyze the drift between adjacent policies to control the error arising from the policy evaluation step.

3. Apart from the regret guarantee, it also provides a PAC guarantee as sample complexity, which allows fair comparison with existing works in this line.

This work is well-organized and clearly articulates each part with the corresponding motivation, challenges, as well as technical novelties in its solutions. Overall, this paper is technically sound and demonstrates considerable technical novelties. It provides a better understanding of PPO in a class of MDPs with function approximation, which potentially benefits policy optimization in practice.


**Weaknesses:**

While this paper provides insights into the PPO algorithm in linear MDPs, it does have several drawbacks:
1. One of the claimed novelties is the multi-batched updating mechanism, which coincides with the similar idea of "policy switch" in literature. However, there is no discussion of the existing works that involve "policy switch", and thus it is not clear whether the computational efficiency is solely brought by the policy switch scheme.

2. Right now, the regret bound requires the batch size to be O(\sqrt{d^3 K}), with leads to the number of batches being O(\sqrt{K}). But whether this choice of batch size and the number of batches is optimal remains to be unknown. It would be worthwhile to study and discuss the balance between the batch size and the number of batches, the optimal choice, how the balance will affect the regret.

3. Right now, this is solely theory-based work. As PPO is applied widely in practice, it will be beneficial to include simple benchmark empirical studies to demonstrate its effectiveness.

**Questions:**

1. In the introduction, the authors mentioned both Cai et al. and He et al. that study linear mixture MDPs are implemented in a model-based manner, whereas existing theoretical studies in linear MDPs are typically value-based methods. However, from the algorithmic perspective, algorithms in Cai et al. and He et al. also perform regularized least-square approximation on value functions in the policy evaluation step. Do you treat all algorithmics that tries to explicitly learn /approximate the transition model (i.e., \hat{p}) as model-based methods in MDPs with function approximation? Could you explain whether your approach falls into the model-based category? If not, which step makes the difference? Is the multi-batched update the main reason that makes the proposed algorithm more computationally efficient compared to the direct extension of the existing OPPO method to linear MDPs from the algorithmic perspective?

**Limitations:**

This is a theory paper with no potential negative social impact under the discussed context.

---

> ### Author Rebuttal · Authors · 2023-08-08
>
> Thanks for your review and positive feedback. We will try to address your concerns in the following.
>
> **Q1:** One of the claimed novelties is the multi-batched updating mechanism, which coincides with the similar idea of "policy switch" in literature. However, there is no discussion of the existing works that involve "policy switch", and thus it is not clear whether the computational efficiency is solely brought by the policy switch scheme.
>
> **A1:** We have briefly discussed previous works that involve multi-batched updating (low policy switch) in Lines 201-203. The algorithms in these works are value-based and cannot tackle adversarial rewards. Furthermore, the corresponding covering number argument is completely new and crucial for the analysis of policy-based algorithms.
>
> Our algorithm is computationally efficient since we can show the running time is polynomial in all parameters (e.g., $d, H, K$), akin to the computational efficiency showcased in LSVI-UCB [1]. Moreover, compared with existing algorithms that update policies in each episode such as LSVI_UCB, our method enjoys better computational efficiency due to the multi-batched update.
>
> [1] Jin, C., Yang, Z., Wang, Z. and Jordan, M. I. (2020). Provably efficient reinforcement learning with linear function approximation. In Conference on Learning Theory. PMLR.
>
>
> **Q2:** Right now, the regret bound requires the batch size to be O(\sqrt{d^3 K}), with leads to the number of batches being O(\sqrt{K}). But whether this choice of batch size and the number of batches is optimal remains to be unknown. It would be worthwhile to study and discuss the balance between the batch size and the number of batches, the optimal choice, how the balance will affect the regret.
>
> **A2:** Based on our current analysis, the choice of batch size is optimal. If we choose the batch size as $B$ (and ignore the dependence of $d$ and $H$), then
> - By the new self-normalized analysis in Appendix C, we have $\beta = \tilde{\mathcal{O}}(\sqrt{K/B})$, which further implies that the model estimation error is bounded by $\beta \cdot \sum_{k=1}^K\sum_{h=1}^H\sqrt{\phi(x_h^k, a_h^k)(\Lambda_h^k)^{-1}\phi(x_h^k, a_h^k)} \le \tilde{\mathcal{O}}(K/\sqrt{B})$ (cf. Lemma 4.5).
> - By Lemma 4.2, we have the policy optimization error is bounded by $\tilde{\mathcal{O}}(\sqrt{KB})$.
>
> Balancing these two terms, we know the optimal choice of $B$ is $\Theta(\sqrt{K})$. We appreciate your valuable suggestion and intend to clarify this in the revision.
>
> **Q3:** Right now, this is solely theory-based work. As PPO is applied widely in practice, it will be beneficial to include simple benchmark empirical studies to demonstrate its effectiveness.
>
> **A3:** Thanks for your suggestions. We will consider adding some empirical results in the future version.
>
> **Q4:** In the introduction, the authors mentioned both Cai et al. and He et al. that study linear mixture MDPs are implemented in a model-based manner, whereas existing theoretical studies in linear MDPs are typically value-based methods. However, from the algorithmic perspective, algorithms in Cai et al. and He et al. also perform regularized least-square approximation on value functions in the policy evaluation step. Do you treat all algorithmics that tries to explicitly learn /approximate the transition model (i.e., \hat{p}) as model-based methods in MDPs with function approximation? Could you explain whether your approach falls into the model-based category? If not, which step makes the difference? Is the multi-batched update the main reason that makes the proposed algorithm more computationally efficient compared to the direct extension of the existing OPPO method to linear MDPs from the algorithmic perspective?
>
> **A4:** For linear mixture MDPs, the transition kernel takes the form $\mathcal{P}_h(s' \mid s, a) = \psi(s, a, s')^\top \beta_h$ with some $\beta_h \in \mathbb{R}^d$. This means that the model of linear mixture MDPs is characterized by $d$ parameters. The algorithms in Cai et al. and He et al. perform regularized least-square regression to directly estimate the *model parameter* $\beta_h$. In contrast, for linear MDPs where $Q_h^\pi(s, a) = \phi(s, a)^\top \theta_h$, and we use the regularized least-square regression to estimate the value function. Therefore, our algorithm does NOT fall into the model-based category. In fact, the number of model parameters of linear MDPs scale with the number of states (the number of model parameters is $d \cdot |\mathcal{S}|$ since $\mu_h(s) \in \mathbb{R}^d$ for all $s\in\mathcal{S}$), and thus difficult to perform the model-based learning efficiently. This difference is one of the major differences between linear mixture MDPs and linear MDPs.  See also [1] and [2] for more discussions.
>
> We want to emphasize that the existing OPPO method (Cai et al.) is restricted to linear mixture MDPs, and its extension to linear MDPs is highly nontrivial (see Challenge 1 in Section 1.1).  The incorporation of multi-batched updating, along with its corresponding analysis, plays a pivotal role in achieving sample efficiency. Furthermore, if we direct extend the existing OPPO method to linear MDPs from the algorithmic perspective (without any modifications)
> - In terms of statistical efficiency, it is hard to provide a theoretical guarantee due to Challenge 1 in Section 1.1.
> - In terms of computational efficiency, you are right --- the utilization of multi-batched update is the main reason that makes the proposed algorithm more computationally efficient.
>
>
> [1] Jin, C., Yang, Z., Wang, Z. and Jordan, M. I. (2020). Provably efficient reinforcement learning with linear function approximation. In Conference on Learning Theory. PMLR.
>
> [2] Ayoub, A., Jia, Z., Szepesvari, C., Wang, M. and Yang, L. (2020). Model-based reinforcement learning with value-targeted regression. In International Conference on Machine Learning. PMLR.

---

> > ### Comment · Reviewer_c9AM · 2023-08-17
> >
> > Thanks for your detailed reply and the further information to provide a better understanding. I am keeping my score and vote for accept. Good luck.

---

> > > ### Author Response · Authors · 2023-08-18
> > >
> > > Thank you for your review and support. We will further polish our paper according to your valuable suggestion.

---

### Official Review · Reviewer_t7wE · 2023-07-07

**Soundness:** 3 good
**Presentation:** 4 excellent
**Contribution:** 3 good
**Rating:** 7
**Confidence:** 3

**Summary:**

This work resolves the known issue for generalizing the policy-based algorithm proposed in [Cai el al.] for linear mixture MDPs to linear MDPs, by multi-batch updating and a bew covering number argument. The proposed model-free policy optimization algorithm advances the theoretical study of PPO in adversarial linear MDPs with full-information feedback.

**Strengths:**

1 This work resolves the issue in generalizing the policy-based algorithm in [Cai. el al.] for linear mixture MDPs to linear MDPs.

2 Besides stochastic linear MDPs, the proposed algorithm can handle adversarial rewards with full-information feedback.

**Weaknesses:**

1 Full-information feedback instead of bandit feedback is considered.

2 The proposed novel techniques developed in this work are insufficient for the policy-based algorithm to achieve the minimax optimal regret.

**Questions:**

1 in Remark 3.3, the authors claim that they achieve the state-of-the-art regret bound for adversarial linear MDPs with full-information feedback. does it mean their result matches or improves the best existing result in linear MDPs? It would be better to give the reference of the best known exisiting result here.

2 Equipped with the new proposed techniques, is it possible to improve the result by refined analysis?

3 It seems the proposed novel techniques are specialized for policy-based algorithm for linear MDPs. Can the authors comment on the impact of the new techniques? For example, by applying those techniques, there is any existing result that can be improved or any hard problem that now can be solved.

---

> ### Author Rebuttal · Authors · 2023-08-08
>
> Thanks for your review and positive feedback. We will try to address your concerns in the following.
>
>
> **Q1:** In Remark 3.3, the authors claim that they achieve the state-of-the-art regret bound for adversarial linear MDPs with full-information feedback. does it mean their result matches or improves the best existing result in linear MDPs? It would be better to give the reference of the best known exisiting result here.
>
>
> **A1:** To the best of our knowledge, there is no previous work that mainly focuses on the adversarial linear MDPs with full-information feedback. However, there are two recent works ([1] and [2]) studying the more challenging adversarial linear MDPs with bandit feedback and their results can directly imply the $\tilde{\mathcal{O}}(d^{2/3}A^{1/9}H^{20/9}K^{8/9})$ and $\tilde{\mathcal{O}}(dH^{2}K^{6/7})$ regret. We have discussed these two related works in the introduction and related works. We will add more discussions in Remark 3.3. Thanks for your suggestion.
>
>
> [1] Dai, Y., Luo, H., Wei, C.-Y. and Zimmert, J. (2023). Refined regret for adversarial mdps with linear function approximation. arXiv preprint arXiv:2301.12942
>
> [2] Sherman, U., Koren, T. and Mansour, Y. (2023). Improved regret for efficient online reinforcement learning with linear function approximation. arXiv preprint arXiv:2301.13087
>
> **Q2:** Equipped with the new proposed techniques, is it possible to improve the result by refined analysis?
>
>
> **A2:**  We think it is possible to improve the result by refined analysis. For instance, if we only perform the policy updating rule $\log K$ times by the doubling trick on the covariance matrix, the model estimation error is at the order of $\tilde{\mathcal{O}}(\sqrt{K})$. However, based on our current analysis, the policy optimization error term will be linear in $K$. If we can make a refined analysis for this term, then we can derive a $\tilde{\mathcal{O}}(\sqrt{K})$ regret as desired. To progress towards a minimax regret bound, techniques in [3] and [4] might be helpful. Thanks for your question, and we intend to delve deeper into these challenges in our future explorations.
>
> [3] Agarwal, A., Jin, Y. and Zhang, T. (2022). Vo q l: Towards optimal regret in model-free rl with nonlinear function approximation. arXiv preprint arXiv:2212.06069.
>
> [4] He, J., Zhao, H., Zhou, D. and Gu, Q. (2022). Nearly minimax optimal reinforcement learning for linear markov decision processes. arXiv preprint arXiv:2212.06132.
>
> **Q3:** It seems the proposed novel techniques are specialized for policy-based algorithm for linear MDPs. Can the authors comment on the impact of the new techniques? For example, by applying those techniques, there is any existing result that can be improved or any hard problem that now can be solved.
>
> **A3:** Based on our new techniques, we improve existing regret bound for adversarial linear MDPs with full-information feedback. Also, we achieve the SOTA regret bound compared with existing policy-based algorithms for stochastic linear MDPs. In our view, these two problems are important and hard. Moreover, our techniques may have several potential applications:
> - Application to multi-agent RL:
>   - In linear Markov games, the covering number issue of value function class still exists and even more severe since the Nash equilibrium maybe stochastic. Our algorithm design and accompanying covering number analysis offer a potential avenue for understanding policy optimization in unknown Markov games. Here we remark that previous works (e.g., [5]) mainly study policy optimization in Markov games with known transitions and rewards.
>   - The statistical hardness of learning Markov games with adversarial opponents is well-established [6]. Our smoothness analysis could potentially shed light on this complex challenge, especially under the assumption of "smooth" policy changes by the adversary.
> - Application to adversarial decision making:
>   - Our algorithm design (especially policy evaluation via average rewards) seems new and may motivate more adversarial decision making problems and algorithms. For example, an interesting finding is that our algorithm does not need to know reward functions at the end of each episode (i.e., full-information feedback). Instead, OPPO+ only requires the average reward function at the end of each batch. This paves the way for tackling a novel class of adversarial decision making problems, where the learner interacts with an adversary and receives the average reward function (or the sum of reward functions) after a set number of episodes (e.g., 100 episodes). Finding more motivational examples of this type of problem and providing more efficient algorithms will be interesting.
>   - Notably, our algorithm is a low switching (multi-batched updating) algorithm for adversarial linear MDPs. We hope algorithm design and analysis can motivate further understanding of low switching algorithms in adversarial decision making.
>
>
> [5] Policy Optimization for Markov Games: Unified Framework and Faster Convergence. Runyu Zhang, Qinghua Liu, Huan Wang, Caiming Xiong, Na Li, Yu Bai
>
>
> [6] Learning Markov Games with Adversarial Opponents: Efficient Algorithms and Fundamental Limits. Qinghua Liu, Yuanhao Wang, Chi Jin.
>
>
> **Q4:** Weakness: (i) Full-information feedback instead of bandit feedback is considered; and (ii) the regret bound is not minimax optimal.
>
> **A4:** Yes, you are right. As we have discussed in our paper, our work cannot tackle these two challenges. These are important open questions and we hope these problems can be addressed in future work. Thanks for your questions.

---

> > ### Comment · Reviewer_t7wE · 2023-08-16
> >
> > Thanks for the detailed response and I am keeping my score.

---

> > > ### Author Response · Authors · 2023-08-17
> > >
> > > Thank you for your review and support. We will further polish our paper according to your valuable suggestion.

---

### Decision · Program_Chairs · 2023-09-21

**Decision:**

Accept (poster)

**Comment:**

This work provides a convergence guarantee for a model-free policy optimization algorithm. The main focus of this work is the linear MDP with full-information and adversarial reward setting(i.e., the reward function is known to the algorithm at the end of each episode). The authors established a novel upper bound guarantee of the order of poly(d,H)K^{3/4}, previously unknown for this setting. this work may enable further progress in this setting (either improving the K^{3/4} convergence or removing the full-information assumption). The reviewers were in consensus for the acceptance of this work and I support this decision.

I further encourage the authors to elaborate on two topics: (1) Strictly speaking, the algorithm the authors analyze resembles more to TRPO, or a mirror-descent like approach to policy optimization. This is different than the standard update role of PPO which relies on clipping and importance sampling ratio. Even though this became standard within the theoretical community, I believe it can be useful to emphasize this discrepancy.
(2) Further highlight the lower bound for this setting and the gap between the lower and upper bound.